# Learning Repetition-Invariant Representations for Polymer Informatics

**Yihan Zhu**
University of Notre Dame
yzhu25@nd.edu

**Gang Liu**
University of Notre Dame
gliu7@nd.edu

**Eric Inae**
University of Notre Dame
einae@nd.edu

**Tengfei Luo**
University of Notre Dame
tluo@nd.edu

**Meng Jiang**
University of Notre Dame
mjiang2@nd.edu

## Abstract

Polymers are large macromolecules composed of repeating structural units known as monomers and are widely applied in fields such as energy storage, construction, medicine, and aerospace. However, existing graph neural network methods, though effective for small molecules, only model the single unit of polymers and fail to produce consistent vector representations for the true polymer structure with varying numbers of units. To address this challenge, we introduce Graph Repetition Invariance (GRIN), a novel method to learn polymer representations that are invariant to the number of repeating units in their graph representations. GRIN integrates a graph-based maximum spanning tree alignment with repeat-unit augmentation to ensure structural consistency. We provide theoretical guarantees for repetition-invariance from both model and data perspectives, demonstrating that three repeating units are the minimal augmentation required for optimal invariant representation learning. GRIN outperforms state-of-the-art baselines on both homopolymer and copolymer benchmarks, learning stable, repetition-invariant representations that generalize effectively to polymer chains of unseen sizes.

## 1 Introduction

Polymers are materials composed of macromolecules made up of multiple repeating units (RUs), i.e., monomers with polymerization points. For example, a homopolymer consists of a single RU repeated along the chain (e.g., polyethylene), resulting in uniform properties. A copolymer interleaves two or more distinct RUs (e.g., styrene-butadiene rubber) to combine or tune material characteristics [5]. Figure 1 illustrates two examples of polymers, which can be encoded as graphs of atoms and bonds. Polymer informatics is an emerging discipline at the intersection of materials science and machine learning, aiming to accelerate the discovery and design of new polymers through data-driven approaches [2, 6, 18, 23, 19], such as graph representation learning. In this direction, graph neural networks (GNNs) have been developed for small-molecule tasks [4, 13, 34, 36]; however, their generalization to polymers has been limited to a single repeating unit.

Ideally, different graph representations of the same polymer should yield identical or highly similar feature vectors. Effective polymer modeling therefore requires learning representations that capture the underlying chemistry of repeating units while remaining invariant to repeat size (i.e., the number of repeats). A similar challenge arises in sentiment analysis, where repeating adjectives, clauses, or sentences should not change the overall sentiment of a text. Recent advances in NLP enforce such repetition invariance to promote stable behaviors of language models [26, 27, 33]. Likewise, robust graph representations for polymer informatics should be expected to exhibit similar invariance.

39th Conference on Neural Information Processing Systems (NeurIPS 2025).

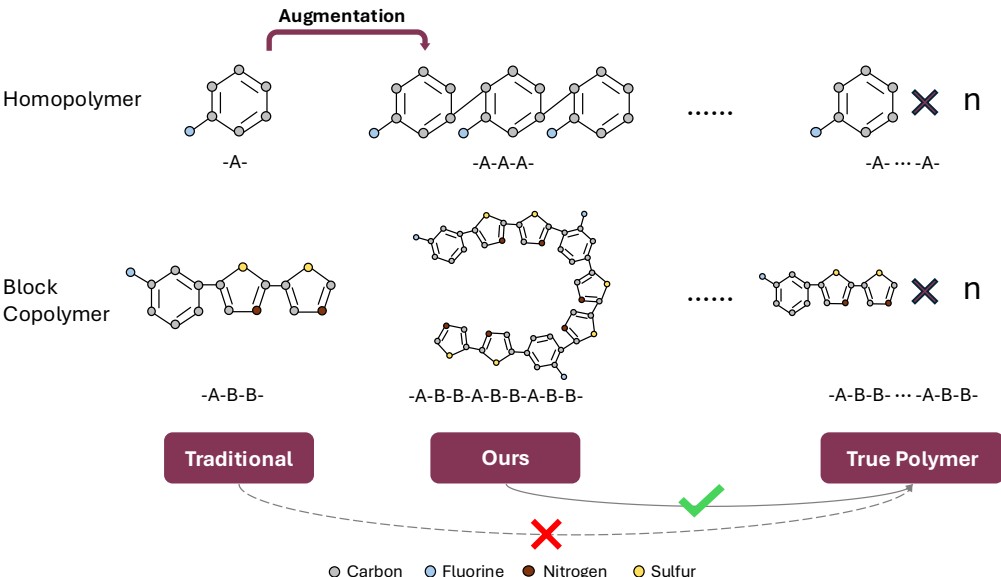

Figure 1: Graph representations of homopolymers and block copolymers. Left and right: Prior graph learning methods model polymers as small molecules using a single repeat unit (e.g., `-A-` or `-A-B-B-`), which may not capture the long-chain features of polymers. Middle and right: Repeating the unit multiple times better approximates realistic polymer structures and serves as an effective data augmentation strategy.

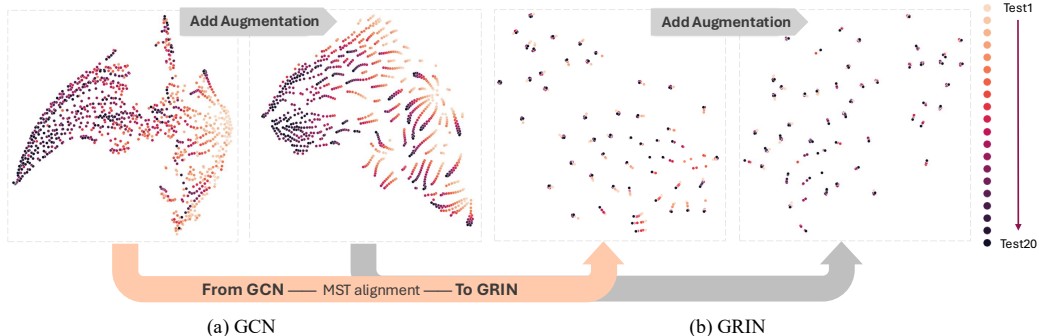

Figure 2: Figure 2: t-SNE visualization of polymer embeddings from GCN and GRIN (ours) across repeat sizes (1–20 RUs) on the glass transition task. Points are colored by repeat count (light=1RU, dark=20RU). (a) GCN produces inconsistent embeddings for different repeat sizes of the same polymer, clustering by size (same color) rather than identity; our augmentation introduces light-to-dark stripes, indicating improved alignment of repeat variants. (b) GRIN learns repeat-invariant representations: each polymer's variants form tight, size-independent clusters, further improved with augmentation.

Figure 2 (a) visualizes the t-SNE embeddings generated by a standard two-layer GCN, trained with and without augmentation, evaluated on test sets augmented with up to 20 RUs. For the same polymer, embeddings, where darker colors indicate larger repeat sizes in its graphs, drift across the latent space as the number of repeats increases. This suggests that conventional message passing entangles repeat size with the core chemical structure of the repeating unit, failing to produce repetition-invariant representations. GCN benefits from repeat-unit augmentation during training, as embeddings of the same polymer with smaller repeat sizes begin to coalesce into visible striations. However, the clusters remain dispersed overall, and embeddings for larger repeat sizes are still scattered, indicating that augmentation alone is not efficient in achieving true generalization.

Accurate property prediction across different graph representations of the same polymer requires *mapping all repeat-size variants to identical or highly similar latent vectors*, while *preserving the*

*chemical semantics in the essential structure*. This joint objective is challenging, as repeat size scales the graph linearly, disrupting the fine-grained information needed for precise representation learning.

In this work, we propose **G**raph **R**epetition **IN**variance (GRIN), a method that combines algorithm alignment with repeat-unit augmentation to address the challenge of repetition-invariant representation learning. GRIN draws inspiration from the edge-greedy logic from the maximum spanning tree (MST) algorithm: dynamic programming(DP)-style updates enforce a fixed reasoning procedure, enabling intrinsic generalization across graph sizes. The MST criterion further preserves the most informative backbone of the polymer graph. We provide theory guarantees for invariant representation learning both model-wise and data-wise: **Model-wise**, GRIN's max-aggregation with a sparsity penalty heuristically encourages alignment with the MST construction process, inheriting the size-generalization guarantees from DP. **Data-wise**, under the repeating unit contraction (definition 3.1), we formally establish latent repetition-invariance (proposition 3.2) and identify the minimal repeat size required for optimal invariant learning (proposition 3.3).

We conduct experiments on four homopolymer and two copolymer datasets. Whereas homopolymer tasks mainly test repeat-size extrapolation, the copolymer datasets probe a harder regime: capturing the synergistic interactions between distinct repeating units. GRIN consistently surpasses strong baselines, e.g., improving homopolymer Density $R^2$ by 10-15% and reducing copolymer Ionization Potential RMSE by 24-30%, while maintaining uniform accuracy across all repeat sizes.

## 2 Related Work

### 2.1 Invariant Representation Learning

Current invariant graph representation learning focuses on identifying features that remain stable across different environments to achieve out-of-distribution (OOD) generalization on graphs [8, 16, 20]. Early work extended invariant risk minimization to graphs, seeking features that remain predictive across training environments [3]. DIR [35] masks environment-specific subgraphs via counterfactual interventions, while Liu et al. [18] proposed a method GREA to learn graph rationales with environment-based augmentations by swapping the environment subgraphs between different samples. Although effective against covariate and local-structure shifts, these approaches largely ignore distributional changes in graph scale.

Size generalization, a specific instantiation of OOD generalization, aiming at learning consistent representations when node or edge counts vary. To this end, SizeShiftReg (SSR) [7] enforces invariance by coarsening graphs during training and aligning embeddings of original and reduced-scale counterparts. Huang et al. [14] proposed a disentanglement loss that separates size-related factors from task-specific features. However, our method explicitly targets the unique challenge of polymers, where key structural information must be preserved as graph size changes due to varying numbers of repeating units.

### 2.2 Neural Algorithmic Alignment

Neural algorithmic alignment has recently emerged as a powerful tool for enhancing GNN size extrapolation [9, 11, 30]. In classical algorithms like Bellman-Ford [10] and Prim [24], solutions are typically constructed through a series of iterations. Neural algorithmic alignment enforces each message-passing layer to mirror one iteration of a known graph algorithm, rather than treating the network as a black-box end-to-end mapper. Although this approach yields impressive OOD performance on well-defined algorithmic benchmarks, its scope remains limited. For instance, Nerem et al. [22] demonstrated provable extrapolation for shortest-path problem by aligning a GNN with Bellman–Ford. Existing models remain restricted to synthetic benchmarks [31] and do not tackle real-world graph classification or regression tasks.

## 3 Methodology

In this section we present Graph Repetition Invariance (GRIN), a novel framework that **i)** augments polymer graphs by chaining repeating units and **ii)** aligns message passing with a maximum-spanning-tree (MST) on that chain. GRIN learns representations invariant to the number of repeating units.

## 3.1 Problem Definition

Prior GNN approaches typically represent a polymer as a monomer graph with polymerization points, $G = (V_A, E_A)$, i.e., a single repeating unit `-A-`, where $V_A$ represents atoms and $E_A$ represent bonds. These methods ignore chain-level connectivity [15, 18, 37]. They apply *mean/sum* aggregation on this monomer graph and assume the learned embedding $h_\theta(G)$ will generalize to arbitrary repeat size $n$, which we found fails: **Model-wise**, *mean* erases all size information and *sum* grows unbounded with $n$. **Data-wise**, real polymers include both intra-monomer and inter-monomer connections, while the previous monomer graph captures only the former.

To capture inter-monomer connectivity, we equip each RU with two anchor vertices $v_i^{\text{in}}, v_i^{\text{out}}$ (the polymerization points, denoted by $*$) and chain them to construct polymer graphs with different number of RUs. Formally, the polymer graph with $n$ repeats is defined as $G^{(n)} = (V, E)$:

$$V = \bigcup_{i=1}^{n} V_A^{(i)}, \qquad E = \Big( \bigcup_{i=1}^{n} E_A^{(i)} \Big) \cup \Big( \bigcup_{i=1}^{n-1} \big\{ (v_{(i)}^{\text{out}}, v_{(i+1)}^{\text{in}}) \big\} \Big).$$

All graphs in the family $\{G^{(n)}\}_{n \in \mathcal{N}}$, share the same RU and an identical ground-truth property value. Training on $\{G^{(n)}\}_{n \in \mathcal{N}}$ enforces a learning objective in which the model must predict the same property $y$ regardless of the repeat count.

We propose a repeat-unit augmentation strategy that generates multiple graph instances of a polymer with varying numbers of RUs. Each RU represents either a homopolymer monomer (`-A-`) or a copolymer motif (`-A-B-`). Real polymers often contain a very large number ($10^2$-$10^4$) of such RUs, far beyond the scope of traditional methods can learn. It is also infeasible to directly augment the training set with all possible repeat sizes. Therefore, in Section 3.3.2, we analyze the minimal repeat size required to learn invariant representations for this augmentation strategy.

## 3.2 Message Passing with Max Aggregation

Given a polymer graph $G = (V, E)$ with initial node features $\{x_v\}_{v \in V}$ and bond features $e_{uv}$, GRIN employs a max-aggregation GNN with $L$ layers. We follow the notation setting in Gilmer et al. [12]. The feature at node $v$ in layer $\ell$ is denoted as $h_v^{(\ell)}$, where the node representation is updated by:

$$h_v^{(\ell)} = h_v^{(\ell-1)} + U^{(\ell)} \left( h_v^{(\ell-1)}, \max_{u \in \mathcal{N}(v)} M^{(\ell)}(h_u^{(\ell-1)}, e_{uv}) \right), \text{where} \tag{1}$$

$$h_v^{(0)} = x_v, \quad M^{(\ell)} : \mathbb{R}^d \times \mathcal{E} \to \mathbb{R}^d, \quad U^{(\ell)} : \mathbb{R}^d \times \mathbb{R}^d \to \mathbb{R}^d.$$

$U^{(\ell)}$ and $M^{(\ell)}$ are multi-layer perceptrons (MLPs) and $\max$ is element-wise maximum over incoming messages. With previous state $h^{l-1}$, any non-max neighbour contributes zero to $U^l$ leaves corresponding feature dimensions remain unchanged. The $M^{(\ell)}$ function takes the features of neighbor $u$ and the edge $(u, v)$ into a message vector. $U^{(\ell)}$ updates the embedding of node $v$. Max aggregation imposes a selection bias toward the strongest neighbor, we add an L1 sparsity penalty on the message and update networks to suppress non-dominant pathways (Eq. 3), encouraging a sparse, MST-like backbone.

## 3.3 Theoretical Guarantee of Invariant Representation Learning

In this section, we provide GRIN's theoretical guarantees for repetition-invariant learning. First, we show that with max aggregation and sparsity constraint, GRIN emulates a greedy DP recurrence and inherits its size-generalization properties. Second, we prove that augmenting with three RUs is sufficient to reach optimal repetition invariance.

### 3.3.1 Model-Wise: Algorithm Alignment with Maximum-Spanning-Tree

For Prim's algorithm [24], let $x_v^{(\ell)} \in \{0, 1\}$ indicate whether node $v$ has been added to the tree after $\ell$ steps. Choose a start node $s$, the MST is constrcuted as

$$x_v^{(1)} = \begin{cases} 1, & v = s, \\ 0, & v \neq s, \end{cases} \quad a_v^{(\ell)} = \max_{u : x_u^{(\ell)} = 1} w_{uv}, \quad x_v^{(\ell+1)} = x_v^{(\ell)} \vee \mathbf{1}[v = \arg \max_{v : x_v^{(\ell)} = 0} a_v^{(\ell)}] \tag{2}$$

where $w$ represents the edge weight, given our message passing design in Eq. (1), we have

$$\underbrace{\max_{u \in \mathcal{N}(v)} M^{(\ell)}\big(h_u^{(\ell-1)}, e_{uv}\big)}_{\text{GNN max-aggregation}} \quad \longleftrightarrow \quad \underbrace{\max_{u \,:\, x_u^{(\ell)}=1} w_{uv}}_{a_v^{(\ell)}}.$$

The additive residual term $h_v^{(\ell-1)}$ in Eq. (1) ensures that only the feature selected by max-aggregation at layer $\ell$ receives an update, while all others satisfy $h_v^{(\ell)} = h_v^{(\ell-1)}$, mirroring Prim's algorithm where nodes not chosen in the greedy step remain unchanged. Additionally, we incorporate an $\ell_1$ sparsity penalty on model parameters as

$$\mathcal{L}_{\text{total}} = \mathcal{L}_{\text{task}} + \lambda \sum_{\ell=1}^{L} (\|\theta_M^{(\ell)}\|_1 + \|\theta_U^{(\ell)}\|_1), \tag{3}$$

encourages many entries of $\theta_M^{(\ell)}, \theta_U^{(\ell)}$ to become exactly zero. This further reinforces the MST alignment by pruning weak connections in the learned message scores, so that only the strongest (MST-relevant) edges survive, enhancing both sparsity and interpretability.

**Max vs. Mean, Sum.** While max-aggregation selects the single most informative incoming message at each node, classical sum- and mean-aggregation mix all neighbor messages[17]. In particular, for sum and mean we have:

$$\textbf{Sum}: h_v^{(\ell)} = U^{(\ell)}\Big(h_v^{(\ell-1)}, \sum_{u \in \mathcal{N}(v)} M^{(\ell)}(h_u^{(\ell-1)}, e_{uv})\Big),$$

$$\textbf{Mean}: h_v^{(\ell)} = U^{(\ell)}\Big(h_v^{(\ell-1)}, \frac{1}{|\mathcal{N}(v)|} \sum_{u \in \mathcal{N}(v)} M^{(\ell)}(h_u^{(\ell-1)}, e_{uv})\Big).$$

Sum-aggregation grows linearly with the node degree, and mean-aggregation normalizes by degree but still blends every neighbor's contribution. Both operations depend on the size of the neighborhood:

$$\sum_{u \in \mathcal{N}(v)} M \quad \propto \quad |\mathcal{N}(v)|, \qquad \frac{1}{|\mathcal{N}(v)|} \sum_{u \in \mathcal{N}(v)} M \quad \text{dilutes extremes as } |\mathcal{N}(v)| \text{ grows.}$$

These findings are further corroborated by the experimental results in Table 12, which show that sum- and mean-aggregation degrade substantially in performance as the repeat size increases. In contrast, by combining max-aggregation with an $\ell_1$ sparsity penalty, GRIN effectively emulates MST construction and thus inherits its inherent size-generalization capabilities, as aligning to DP recurrences enjoy out-of-distribution generalization across input scales [38].

### 3.3.2 Data-Wise: Minimal Repeat Size for Invariant Representation

One can repeat the polymer unit (-A-) as the augmented data point $P_n$, it can be viewed as an undirected graph $G = (V_G, E_G)$. Let $n := |V_G|/|V_A|$ represents the number of repeats (or the degree of polymerization). We analyze the model behavior over an abstract hyperchain level and show that repeat-augmented polymers $P_m$ ($m \geq 2$), the model can apply same update rule at each layer, yielding a constant output $y*$, irrespective of the repeat size.

**Definition 3.1** (Polymer Hyperchain). We contract each repeating unit -A- in the polymer graph to a single *supernode* and neglecting all intra-monomer edges, while retaining only the inter-monomer polymerization edges. $P_n$ can be abstracted as a hyperchain

$$S_n = \{s_1, s_2, \ldots, s_n\}, \ \ E_n = \big\{\{s_i, s_{i+1}\} \mid i = 1, \ldots, n-1\big\}.$$

This definition based on the dominant backbone of a polymer chain and enforces a one-dimensional DP structure for message passing as Dudzik and Veličković [9]. For each supernode $s_i \in S_n$, we define its neighborhood and **hyperdegree** as

$$\mathcal{N}(s_i) := \{s_j \mid \{s_i, s_j\} \in E\}, \quad \deg_P(s_i) := \big|\mathcal{N}(s_i)\big| = \begin{cases} 0, & n = 1, \\ 1, & i = 1 \text{ or } i = n, \\ 2, & i = 2, \ldots, n-1 \text{ for } n > 2. \end{cases} \tag{4}$$

In the following analysis, hyperdegree-n and $\deg_P(\cdot) = n$ denote supernodes with $n$ hyperdegree.

**Hyperchain Message Passing.** We denote by $h_s^{(t)} \in \mathbb{R}^d$ the embedding of supernode $s$ after the $t$-th layer, and $m_s^{(t)}$ as its aggregated message. Then layer-wise message passing reduces to

$$m_{s_i}^{(t)} = \begin{cases} 0, & \deg_P(\cdot) = 0, \\ \max_{s_j \in \mathcal{N}(s_i)} M_\theta\big(h_{s_j}^{(t)}, e_{j,i}\big), & \deg_P(\cdot) \geq 1, \end{cases} \tag{5}$$

$$h_{s_i}^{(t+1)} = U_\theta\big(h_{s_i}^{(t)}, m_{s_i}^{(t)}\big). \tag{6}$$

Given Eq. (5), we notice that training only on $P_1$ (hyperchain of length 1) activates solely the $\deg_P(\cdot) = 0$ branch, leaving $\deg_P(\cdot) \geq 1$ parameters unlearned which is essential in extrapolating to longer chains. To address this issue, we supervise on a merge set $\{P_1, P_n\}$ with a shared target $y^\star$. Let loss be written as:

$$\mathcal{L}(\theta) = \big\|h_{s_i}^{(T)}(P_1; \theta) - y^\star\big\|_2^2 + \big\|h_{s_i}^{(T)}(P_n; \theta) - y^\star\big\|_2^2 + \lambda\|\theta\|_1.$$

Here $h_{s_i}^{(t)}$ is supernode $s_i$'s embedding after $t$ layers, and $\|\theta\|_1$ is the $\ell_1$ term introduced in Eq. (3).

**Proposition 3.2** (Latent Repetition-Invariance). *Under definition 3.1 and let $\theta^\star \in \arg\min \mathcal{L}(\theta)$, for every test hyperchain $P_m$ with $m \geq 2$, the prediction is*

$$f_{\theta^\star}\big(P_m\big) = y^\star.$$

Training on $\{P_1, P_n\}$ exposes both structural cases—nodes of $\deg_P(\cdot) = 0$ and $\deg_P(\cdot) \geq 1$—and the $\ell_1$ term zeros out all other pathways, which means the complete update rule in Eq. (5) is learned.

**Hyperdegree-2 Supervision** Introducing augmentation $P_n$ with supernodes satisfying $\deg_P(\cdot) \geq 1$ helps model learn extrapolation across repeating units. A hyperdegree-1 supernode receives exactly one incoming message, so its gradient flows along a single branch. In contrast, a hyperdegree-2 supernode has two competing branches, introducing supervision for multi-branch aggregation.

Let $\delta_s = \partial\mathcal{L}/\partial m_s \in \mathbb{R}^d$ be the back-propagated error arriving at a hyperdegree-2 supernode $s$ with two neighbors $s_\ell, s_r$. Under classical GNN contraction assumption[28], each message–passing layer is assumed to be $L$-Lipschitz, i.e., it satisfies $|f(x) - f(x')| \leq L|x - x'|$ for all inputs $x$, $x'$ and some constant $L < 1$, so any gradient norm is reduced by at most a factor of $L$ per layer.

**Proposition 3.3** (Accumulated Gradient Norm). *When supervising on $P_n$ with $n \geq 3$, the total back-propagated gradient norm at any degree-2 supernode satisfies:*

$$\|\nabla_{\mathcal{L}}^{(3)}\|_2 = \|\delta\|_2 L, \quad \|\nabla_{\mathcal{L}}^{(n)}\|_2 = \|\delta\|_2 L \frac{1 - L^{n-2}}{1 - L}, \quad \|\delta\|_2 = \max_s \|\delta_s\|_2.$$

Under the hyperchain abstraction, $P_3$ (the smallest sample containing a hyperdegree-2 node) is the minimal configuration for GRIN to learn true multi-branch fusion. $P_2$ never exposes this dual-message scenario and thus could be insufficient. Furthermore, given the contraction factor $L$, increasing repeat size beyond 3 yields only geometrically vanishing improvements with additional training cost.

## 4 Experiments

We conduct experiments for three research questions:

- **Q1) Effectiveness:** Does GRIN make more accurate predictions for homopolymers and copolymers than existing methods?

- **Q2) Repetition-Invariance Learning:** Does GRIN learn invariant representations and maintain performance across varying repeat sizes?

- **Q3) Ablation study:** What is the minimal repeat size required for augmentation? How does the merge ratio (proportion of augmented graphs) impact performance?

Table 1: Statistics of six datasets for property prediction.

| Dataset | Property | # Graphs | Avg # Size | Max # Size |
|---------|----------|----------|------------|------------|
| Homopolymer | GlassTemp | 7,174 | 20.5 | 69 |
| | MeltingTemp | 3,651 | 18.3 | 60 |
| | PolyDensity | 1,694 | 15.9 | 48 |
| | O2Perm | 595 | 18.0 | 49 |
| Copolymer | EA | 3,000 | 17.7 | 36 |
| | IP | 3,000 | 17.7 | 36 |

## 4.1 Experimental Settings

**Datasets.** We evaluate property prediction task on four homopolymer datasets and two copolymer datasets. The four datasets predict the glass transition temperature (GlassTemp, $^{\circ}$C), polymer density (PolyDensity, g/cm$^3$), melting temperature (MeltingTemp, $^{\circ}$C) and oxygen permeability (O$_2$Perm, Barrer). The other two are about electron affinity (EA, eV) and ionization potential (IP, eV). The dataset statistics are given in Table 1, in which the *Size* refers to the diameter of graph calculated by $\mathrm{dia}(G) = \max_{u,v \in V} d_G(u, v)$. For every polymer, we construct a family of polymer graphs $\{G^{(i)}\}_{i=1}^n$ as Section 3.1. We use **GRIN-RepAug** to denote model training on $G^{(1)}$ and **GRIN** for model training with repeat-unit augmentation $\{G^{(1)}, G^{(3)}\}$. Dataset details can be found in the appendix A.

**Evaluation and Baseline.** We evaluate the regression performance using the coefficient of determination (R$^2$) and Root Mean Squared Error (RMSE). For baselines, we compare our GRIN with methods designed specifically for size-generalization: DISGEN, SSR, BFGNN [7, 14, 22] and others for general OOD, including GREA, DIR, IRM, RPGNN [3, 18, 21, 35]. We test both GIN and GCN as graph encoder for all models. Please refer to appendix B for details of implementation and appendix C for computational efficiency comparison.

## 4.2 Results on Effectiveness (Q1) and Repetition-Invariant Representation (Q2)

We evaluate on $\{G^{(i)}\}_{i=1}^{60}$ for homopolymer and $\{G^{(i)}\}_{i=1}^{20}$ for copolymer, graph diameter ranges from 10 to 10$^4$. GRIN consistently achieves best results among all tasks, GRIN-RepAug ranks the second, which demonstrate the method's effectiveness of repetition-invariant learning. All observations hold for both GCN and GIN backbones, confirming that the gain stems from the repeat-aware alignment itself rather than the choice of encoder. More results can be found at appendix D.

**Polymer Datasets** Table 2 and Table 3 report R$^2$ and RMSE on Test1 (1 RU) and Test60 (60 RUs). Test1 measures property prediction performance, whereas Test60 probes a 60× repeat-size extrapolation.

- **Effectiveness**: Across all four homopolymer tasks, GRIN with/without augmentation achieves top-2 accuracy on all test sets. Compared to the strongest baseline, GRIN achieves a 1-2% improvement on GlassTemp, MeltingTemp, and O$_2$Perm, and a remarkable 15% on PolyDensity. The advantage widens under repeat-size extrapolation: the improvement exceeds 10% for Test60. Several baselines collapse on Test60 (negative R$^2$). In particular, DISGEN almost fails on PolyDensity and O$_2$Perm: its augmentation pipeline depends on a GNN-Explainer module that can misprioritize nodes (common in the noisy, low-sample O$_2$Perm and PolyDensity datasets), results in a training failure.

- **Repetition Invariance**: GRIN's performance gap between Test1 and Test60 stays within ±3%, confirming that the model indeed learns repetition-invariant representation. Once trained on 1-RU and 3-RU graphs, it transfers to polymers sixty times larger without loss of accuracy.

**Copolymer Datasets** Table 4 reports R$^2$ and RMSE on Test1 (1 RU) and Test20 (20 RUs). Test1 measures property prediction performance, whereas Test20 probes a 20× repeat-size extrapolation.

Table 2: Results on homopolymer datasets (GlassTemp, MeltingTemp) with 1 repeating unit (Test1) and 60 repeating units (Test60): **GRIN consistently achieves the highest R2 and smallest RMSE.**

| | Model | GlassTemp | | | | MeltingTemp | | | |
|---|---|---|---|---|---|---|---|---|---|
| | | Test1 | | Test60 | | Test1 | | Test60 | |
| | | $R^2\uparrow$ | RMSE$\downarrow$ | $R^2\uparrow$ | RMSE$\downarrow$ | $R^2\uparrow$ | RMSE$\downarrow$ | $R^2\uparrow$ | RMSE$\downarrow$ |
| GCN | GCN [15] | 0.878±0.001 | 38.7±0.1 | 0.818±0.009 | 47.2±1.0 | 0.693±0.002 | 62.6±0.2 | 0.664±0.004 | 65.5±0.4 |
| | DIR [35] | 0.701±0.054 | 60.8±5.7 | < 0 | > 100 | 0.374±0.235 | 88.2±14.4 | < 0 | > 100 |
| | GREA [18] | 0.870±0.005 | 40.0±0.8 | < 0 | > 100 | 0.700±0.005 | 61.9±0.5 | < 0 | > 100 |
| | IRM [3] | 0.872±0.016 | 39.7±2.5 | < 0 | > 100 | 0.696±0.010 | 62.3±1.0 | < 0 | > 100 |
| | RPGNN [21] | 0.888±0.005 | 37.1±0.8 | < 0 | > 100 | 0.691±0.017 | 62.8±1.7 | < 0 | > 100 |
| | SSR [7] | 0.772±0.044 | 52.8±5.2 | < 0 | > 100 | 0.693±0.027 | 62.6±2.8 | < 0 | > 100 |
| | DISGEN [14] | 0.885±0.005 | 37.6±0.8 | 0.846±0.004 | 43.5±0.6 | 0.723±0.003 | 60.1±0.3 | 0.653±0.013 | 66.6±1.3 |
| | BFGCN [22] | 0.889±0.003 | 37.0±0.5 | 0.850±0.016 | 43.0±2.2 | 0.698±0.006 | 62.1±0.6 | 0.683±0.013 | 63.6±1.3 |
| | **GRIN-RepAug** | 0.890±0.001 | 36.8±0.1 | 0.866±0.014 | 40.6±2.0 | 0.741±0.007 | 57.5±0.8 | 0.707±0.009 | 61.1±0.9 |
| | **GRIN** | **0.893±0.001** | **36.3±0.2** | **0.892±0.001** | **36.5±0.2** | **0.745±0.004** | **57.0±0.4** | **0.746±0.002** | **56.9±0.2** |
| GIN | GIN [37] | 0.882±0.003 | 38.1±0.5 | 0.848±0.001 | 43.3±0.1 | 0.697±0.007 | 62.2±0.7 | 0.668±0.015 | 65.1±1.5 |
| | DIR [35] | 0.600±0.087 | 70.1±6.0 | < 0 | > 100 | 0.270±0.124 | 95.9±8.1 | < 0 | > 100 |
| | GREA [18] | 0.865±0.008 | 40.7±1.2 | < 0 | > 100 | 0.705±0.010 | 61.4±1.0 | < 0 | > 100 |
| | IRM [3] | 0.881±0.002 | 38.3±0.3 | < 0 | > 100 | 0.699±0.009 | 62.0±0.9 | < 0 | > 100 |
| | RPGNN [21] | 0.889±0.004 | 37.0±0.7 | < 0 | > 100 | 0.702±0.013 | 61.2±1.3 | < 0 | > 100 |
| | SSR [7] | 0.834±0.053 | 44.9±7.0 | < 0 | > 100 | 0.610±0.138 | 69.9±12.0 | < 0 | > 100 |
| | DISGEN [14] | 0.885±0.002 | 37.6±0.4 | 0.851±0.012 | 42.8±1.8 | 0.726±0.011 | 59.2±1.2 | 0.646±0.023 | 67.2±2.1 |
| | BFGIN [22] | 0.888±0.002 | 37.1±0.4 | 0.842±0.010 | 44.1±1.4 | 0.697±0.006 | 62.0±0.6 | 0.676±0.016 | 64.3±1.6 |
| | **GRIN-RepAug** | 0.894±0.003 | 36.2±0.5 | 0.876±0.009 | 39.0±1.5 | 0.736±0.006 | 57.9±0.6 | 0.705±0.001 | 61.4±0.1 |
| | **GRIN** | **0.896±0.001** | **35.7±0.1** | **0.895±0.001** | **36.0±0.1** | **0.740±0.001** | **57.8±0.1** | **0.739±0.002** | **57.7±0.3** |

Table 3: Results on homopolymer datasets (PolyDensity, $O_2$Perm) with 1 repeating unit (Test1) and 60 repeating units (Test60): **GRIN consistently achieves the highest R2 and smallest RMSE.**

| | Model | PolyDensity | | | | $O_2$Perm | | | |
|---|---|---|---|---|---|---|---|---|---|
| | | Test1 | | Test60 | | Test1 | | Test60 | |
| | | $R^2\uparrow$ | RMSE$\downarrow$ | $R^2\uparrow$ | RMSE$\downarrow$ | $R^2\uparrow$ | RMSE$\downarrow$ | $R^2\uparrow$ | RMSE$\downarrow$ |
| GCN | GCN [15] | 0.681±0.003 | 0.125±0.001 | 0.660±0.035 | 0.129±0.007 | 0.870±0.008 | 786.7±23.7 | 0.876±0.034 | 761.7±110.4 |
| | DIR [35] | 0.662±0.021 | 0.129±0.004 | < 0 | > 1 | 0.121±0.078 | 2045.3±92.4 | < 0 | > 3000 |
| | GREA [18] | 0.715±0.017 | 0.118±0.003 | < 0 | > 1 | 0.921±0.024 | 609.1±98.8 | < 0 | > 3000 |
| | IRM [3] | 0.685±0.010 | 0.124±0.002 | < 0 | > 1 | 0.874±0.079 | 752.7±230.6 | < 0 | > 3000 |
| | RPGNN [21] | 0.662±0.004 | 0.129±0.001 | < 0 | > 1 | 0.096±0.012 | 2075.2±14.2 | < 0 | > 3000 |
| | SSR [7] | 0.376±0.149 | 0.174±0.021 | < 0 | > 1 | 0.829±0.070 | 887.9±190.8 | < 0 | > 3000 |
| | DISGEN [14] | 0.203±0.104 | 0.197±0.011 | 0.189±0.111 | 0.199±0.014 | < 0 | > 3000 | < 0 | > 3000 |
| | BFGCN [22] | 0.633±0.036 | 0.134±0.006 | 0.631±0.061 | 0.134±0.011 | 0.916±0.014 | 631.5±52.6 | 0.901±0.014 | 685.2±49.1 |
| | **GRIN-RepAug** | 0.720±0.018 | 0.117±0.004 | 0.715±0.031 | 0.118±0.007 | 0.923±0.008 | 604.7±31.9 | 0.910±0.008 | 655.2±28.6 |
| | **GRIN** | **0.730±0.017** | **0.115±0.004** | **0.747±0.009** | **0.111±0.002** | **0.929±0.002** | **583.4±7.5** | **0.929±0.002** | **581.7±7.2** |
| GIN | GIN [37] | 0.691±0.011 | 0.123±0.002 | 0.618±0.036 | 0.137±0.006 | 0.853±0.019 | 836.3±53.4 | 0.865±0.014 | 801.4±40.7 |
| | DIR [35] | 0.619±0.060 | 0.136±0.011 | < 0 | > 1 | 0.517±0.297 | 1432.3±527.1 | < 0 | > 3000 |
| | GREA [18] | 0.723±0.008 | 0.117±0.002 | < 0 | > 1 | 0.918±0.024 | 619.5±94.3 | < 0 | > 3000 |
| | IRM [3] | 0.688±0.008 | 0.124±0.002 | < 0 | > 1 | 0.858±0.091 | 796.3±251.5 | < 0 | > 3000 |
| | RPGNN [21] | 0.629±0.039 | 0.135±0.007 | < 0 | > 1 | 0.096±0.012 | 2075.2±14.2 | < 0 | > 3000 |
| | SSR [7] | 0.448±0.022 | 0.165±0.003 | < 0 | > 1 | 0.745±0.027 | 1101.5±59.8 | < 0 | > 3000 |
| | DISGEN [14] | 0.213±0.053 | 0.196±0.007 | 0.119±0.060 | 0.208±0.007 | < 0 | > 3000 | < 0 | > 3000 |
| | BFGIN [22] | 0.673±0.007 | 0.127±0.001 | 0.678±0.020 | 0.126±0.004 | 0.924±0.010 | 601.6±38.7 | 0.913±0.009 | 643.6±32.3 |
| | **GRIN-RepAug** | 0.715±0.005 | 0.117±0.001 | 0.720±0.013 | 0.117±0.003 | 0.930±0.001 | 577.9±5.3 | 0.916±0.003 | 631.2±11.7 |
| | **GRIN** | **0.731±0.004** | **0.115±0.001** | **0.752±0.006** | **0.110±0.001** | **0.930±0.007** | **577.2±28.5** | **0.929±0.006** | **580.9±24.9** |

- **Effectiveness**: GRIN ranks first in every setting and GRIN-RepAug the second. Against the strongest baseline, GRIN raises $R^2$ by 1.8-3.7% and cuts RMSE by 14-30% on Test1. The margin widens on Test20, reaching 5.2-7.4% ($R^2$) and 24-35% (RMSE). Baselines show the same sharp performance decline on larger graphs as observed in the homopolymer experiments, underscoring the challenge of repeat-size extrapolation.

- **Repetition Invariance**: When graph size grows $20\times$, GRIN's accuracy remains nearly unchanged: $R^2$ drops by at most 0.5% and RMSE increases by less than 2%.

## 4.3 Ablation Study

**Minimal Merge Repeat Size**   In this experiment, we evaluate the impact of different training pairs on MeltingTemp (homopolymer) and IP (copolymer). Figure 3 presents the results. Training pair $\{1, 3\}$ (with 3-RU augmentation) reaches the best improvement. Further increasing the merge size beyond 3 leads to convergence, with no significant additional gains. These observations are highly compatible with our theory in Section 3.3.2: the $\{1, 2\}$ training set lacks binary-update supervision, and the hyperdegree-2 gradient strength plateaus for merge size $n > 3$.

Table 4: Results on copolymer datasets (EA, IP) with 1 repeating unit (Test1) and 20 repeating units (Test20): **GRIN consistently achieves the highest R2 and smallest RMSE**.

| | Model | EA | | | | IP | | | |
|---|---|---|---|---|---|---|---|---|---|
| | | Test1 | | Test20 | | Test1 | | Test20 | |
| | | $R^2\uparrow$ | RMSE$\downarrow$ | $R^2\uparrow$ | RMSE$\downarrow$ | $R^2\uparrow$ | RMSE$\downarrow$ | $R^2\uparrow$ | RMSE$\downarrow$ |
| GCN | GCN [15] | 0.939±0.006 | 0.146±0.008 | 0.887±0.023 | 0.199±0.020 | 0.918±0.004 | 0.141±0.004 | 0.909±0.004 | 0.149±0.003 |
| | DIR [35] | < 0 | > 1 | < 0 | > 1 | < 0 | > 1 | < 0 | > 1 |
| | GREA [18] | 0.939±0.007 | 0.146±0.008 | < 0 | > 1 | 0.905±0.069 | 0.146±0.053 | < 0 | > 1 |
| | IRM [3] | 0.930±0.012 | 0.157±0.013 | < 0 | > 1 | < 0 | > 1 | < 0 | > 1 |
| | RPGNN [21] | 0.929±0.003 | 0.158±0.004 | < 0 | > 1 | 0.914±0.020 | 0.144±0.017 | < 0 | > 1 |
| | SSR [7] | 0.818±0.091 | 0.248±0.062 | < 0 | > 1 | 0.883±0.021 | 0.168±0.015 | < 0 | > 1 |
| | DISGEN [14] | 0.163±0.081 | 0.542±0.027 | 0.179±0.082 | 0.537±0.027 | 0.232±0.047 | 0.432±0.013 | 0.174±0.062 | 0.447±0.017 |
| | BFGCN [22] | 0.904±0.014 | 0.184±0.014 | 0.883±0.018 | 0.203±0.016 | 0.916±0.014 | 0.137±0.006 | 0.901±0.014 | 0.146±0.008 |
| | **GRIN-RepAug** | 0.952±0.005 | 0.129±0.007 | 0.942±0.006 | 0.143±0.008 | 0.946±0.010 | 0.113±0.013 | 0.936±0.010 | 0.125±0.010 |
| | **GRIN** | **0.956±0.002** | **0.125±0.002** | **0.952±0.001** | **0.129±0.002** | **0.952±0.005** | **0.108±0.006** | **0.947±0.005** | **0.113±0.005** |
| GIN | GIN [37] | 0.918±0.006 | 0.170±0.006 | 0.903±0.007 | 0.185±0.007 | 0.925±0.004 | 0.135±0.004 | 0.914±0.004 | 0.144±0.003 |
| | DIR [35] | < 0 | > 1 | < 0 | > 1 | < 0 | > 1 | < 0 | > 1 |
| | GREA [18] | 0.831±0.091 | 0.237±0.072 | < 0 | > 1 | 0.918±0.002 | 0.141±0.002 | < 0 | > 1 |
| | IRM [3] | 0.930±0.010 | 0.157±0.011 | < 0 | > 1 | 0.934±0.003 | 0.126±0.003 | < 0 | > 1 |
| | RPGNN [21] | 0.929±0.001 | 0.158±0.002 | < 0 | > 1 | 0.917±0.025 | 0.141±0.021 | < 0 | > 1 |
| | SSR [7] | 0.810±0.161 | 0.245±0.104 | < 0 | > 1 | 0.860±0.067 | 0.182±0.042 | < 0 | > 1 |
| | DISGEN [14] | 0.123±0.146 | 0.554±0.048 | 0.100±0.107 | 0.565±0.035 | 0.232±0.047 | 0.432±0.013 | 0.181±0.060 | 0.447±0.016 |
| | BFGIN [22] | 0.925±0.006 | 0.163±0.007 | 0.918±0.004 | 0.170±0.004 | 0.929±0.006 | 0.131±0.006 | 0.918±0.002 | 0.141±0.002 |
| | **GRIN-RepAug** | 0.954±0.004 | 0.127±0.006 | 0.947±0.004 | 0.137±0.005 | 0.963±0.001 | 0.095±0.001 | 0.953±0.003 | 0.107±0.004 |
| | **GRIN** | **0.961±0.001** | **0.117±0.002** | **0.960±0.001** | **0.119±0.002** | **0.965±0.001** | **0.092±0.001** | **0.962±0.001** | **0.096±0.001** |

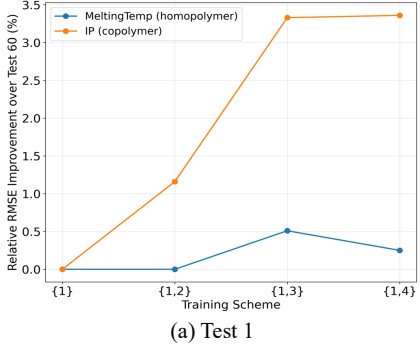
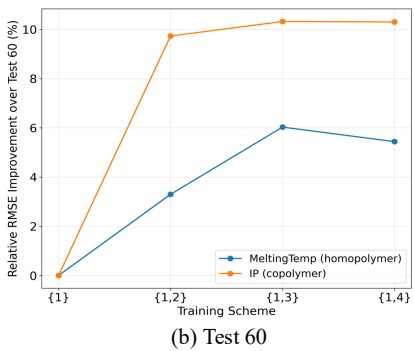

(a) Test 1
(b) Test 60

Figure 3: Performance improvement of GRIN over GRIN-RepAug (without augmentation) under different training schemes (GIN-based). The test sets include 1 repeating unit (a) and 60 repeating units (b) for MeltingTemp (homopolymer) and IP (copolymer), respectively. In both cases, training on the $\{1, 3\}$ pair reaches the best improvement, while further increasing the merging size leads to convergence.

**Effect of Merge Ratio**   We investigate the effect of different merge ratios in repeat-unit augmentation on the training set. Since we have demonstrated both theoretically and empirically that a repeat size of three is optimal, this experiment uses the training set $\{1, 3\}$. Unsurprisingly, we can observe that in Table 5, a balanced 1:1 ratio gives the best performance, while reducing the proportion of augmented samples to 1:0.9 and 1:0.8 leads to a monotonic decline in accuracy. This observation further validates our analysis: weakening binary-update supervision (from hyperdegree-2 supernodes) slightly diminishes training effectiveness. More ablation studies are provided in appendix D.3.

## 4.4   Visualization

Figure 2 visualizes the t-SNE embeddings generated by GCN and GRIN, trained with and without augmentation, evaluated on test sets augmented up to 20 RUs. As discussed in Section 1, for the same polymer, GCN's embeddings remain sensitive to repeat size, even with augmentation. In contrast, as shown in Figure 2 (b), GRIN produces stable embeddings across all test lengths, when combined with 3-RU augmentation, it collapses all repeats of the same polymer into a single cluster. This behavior aligns with the cosine similarity of 0.999 between repeat sizes 1 and 60 reported in Table 11, demonstrating repetition invariance.

Table 5: Merge-Ratio (1RU vs. 3RUs) Ablation of GRIN: **Weakening binary-update supervision (comes from supernodes with hyperdegree = 2) harms performance**. The test sets include 1 RU and 60 RUs.

| | Ratio | MeltingTemp | | | |
| --- | --- | --- | --- | --- | --- |
| | | Test1 | | Test60 | |
| | | $R^2 \uparrow$ | RMSE $\downarrow$ | $R^2 \uparrow$ | RMSE $\downarrow$ |
| GCN | 1:1 | **0.745±0.004** | **57.021±0.417** | **0.746±0.002** | **56.934±0.199** |
| | 1:0.9 | 0.736±0.005 | 58.088±0.491 | 0.734±0.001 | 58.273±0.089 |
| | 1:0.8 | 0.729±0.005 | 58.866±0.496 | 0.729±0.004 | 58.873±0.387 |
| GIN | 1:1 | **0.740±0.001** | **57.809±0.038** | **0.739±0.002** | **57.716±0.257** |
| | 1:0.9 | 0.733±0.009 | 58.443±0.950 | 0.733±0.008 | 58.353±0.866 |
| | 1:0.8 | 0.730±0.004 | 58.759±0.437 | 0.729±0.009 | 58.837±0.949 |

## 5   Conclusion

In this work, we addressed limitations of prior polymer learning methods that modeled polymers as single monomers and thus failed to capture their realistic, repetitive structures. We made the first attempt to achieve repetition-invariant representation learning. To this end, we proposed GRIN, which combined MST-aligned aggregation with a repeat-unit-based data augmentation strategy. We provided a theoretical analysis of the minimal augmentation size required to achieve invariance. Experiments on both polymer and copolymer datasets showed that GRIN learned embeddings that reliably generalized to polymers with long repeat lengths.

## Acknowledgement

This work was partially supported by NSF IIS-2142827, IIS-2146761, IIS-2234058, and CBET-2332270. We also appreciate the support from the Foundation Models and Applications Lab of Lucy Institute and ND-IBM Tech Ethics Lab.

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

# A  Dataset Details

## A.1  Labeled Data

We evaluate our models on four homopolymer property datasets and two copolymer datasets, each split into 60% training, 10% validation, and 30% test sets. To prevent data leakage, we apply repeat-unit augmentation independently within each split: for every dataset, we generate augmented graphs with repeat sizes from 2 to 60. On the training set, we benchmark four augmentation schemes—{1,2}, {1,3}, {1,2,3}, and {1,4}—while the test set is evaluated on graphs spanning all repeat sizes from 1 through 60.

**Homopolymer**  *GlassTemp*, *MeltingTemp*, *PolyDensity*, and *$O_2Perm$*. Targets are predicting glass transition temperature (GlassTemp, °C), polymer density (PolyDensity, $g/cm^3$), melting temperature (MeltingTemp, °C) and oxygen permeability ($O_2$Perm, Barrer), respectively. The first three datasets are extracted from Polymer Info [23], *$O_2Perm$* is compiled from the Membrane Society of Australasia portal following [29].

The distributions displayed in Figure 4 (a)-(d) indicate that GlassTemp ($\sim$7000 samples) and MeltingTemp ($\sim$3600 samples) exhibit only moderate skew, which corresponds to stable performance across all models. PolyDensity ($\sim$1700 samples) is nearly Gaussian but limited to its smaller size, causing some baselines to underperform. $O_2$Perm ($\sim$600 samples) is both heavily right-skewed and long-tailed on a very small dataset, leading most methods to fail on the high-permeability cases.

**Copolymer**  *EA* (Electron Affinity, eV) and *IP* (Ionization Potential, eV). These two datasets conducted from the same SMILES strings with different properties. Data are obtained and processed as Aldeghi and Coley [2]. Figure 4 (e) and (f) show that both properties are well covered and exhibit only moderate skew.

## A.2  Unlabeled Data

We leverage 12,764 unlabeled polymers to further evaluate model generalization by generating graph representations, as summarized in Table 11.

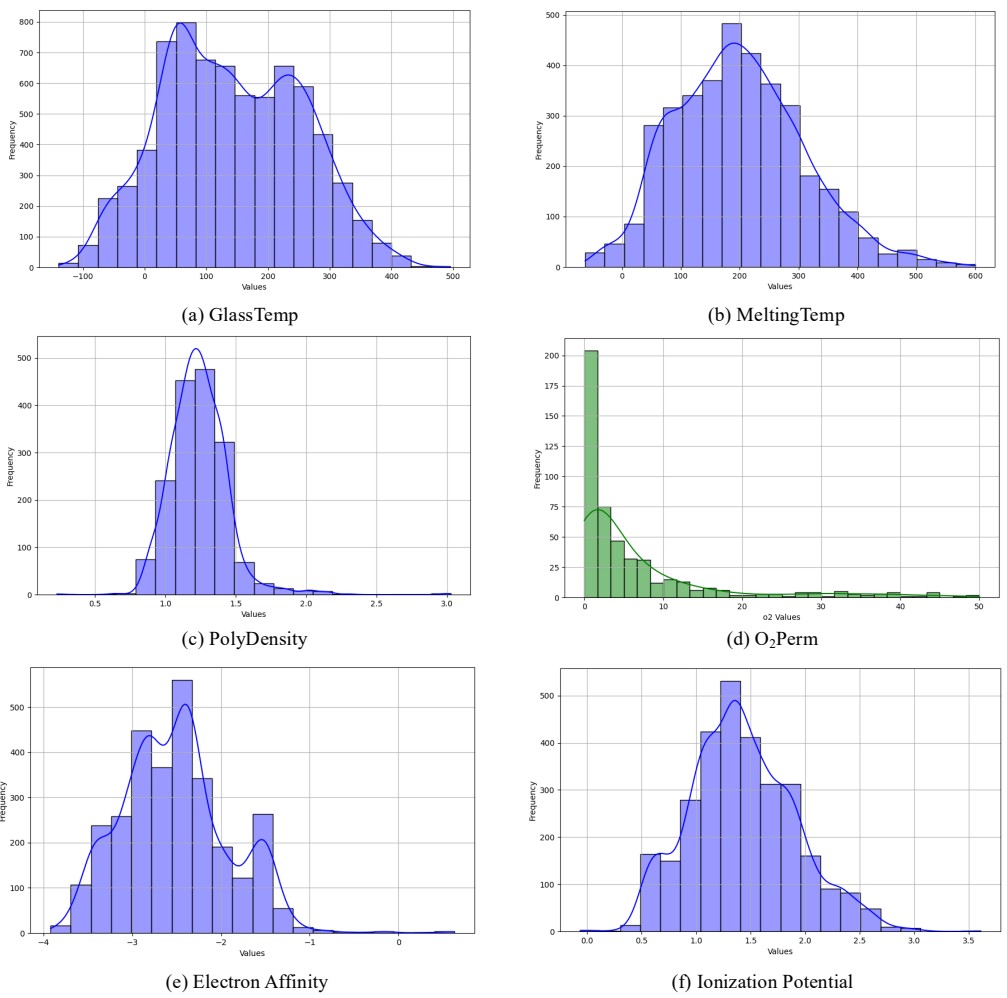

Figure 4: Property distributions of homopolymer datasets (a) GlassTemp, (b) MeltingTemp, (c) PolyDensity, (d) $O_2$Perm and copolymer datasets (e) Electron Affinity, (f) Ionization Potential. The x-axis denotes property value and the y-axis denotes frequency. For clarity, the $O_2$Perm histogram with long tail is truncated to the 0–50 value range.

# B  Implementation Details

All experiments were conducted on a 16-core Intel Xeon Gold 6130 CPU (2.1 GHz) with 96 GB RAM and a single NVIDIA A6000 GPU (48 GB). Code is implemented in PYTHON 3.11 and PYTORCH 2.1.0+cu118, with graph processing using PYG 2.5.1.

**Model configuration.**  Each model is evaluated with two backbone encoders, GIN and GCN. Since both GRIN and BFGNN[22] are based on algorithmic alignment, they share the same hyperparameter configuration for each task.

- Layer Number $\in \{2, 3\}$,
- Learning Rate $\in \{1e\text{-}2, 1e\text{-}3\}$,
- Batch = 32,
- Hidden Dimension = 300,
- $\ell_1$ Regularization Weight = 1e-3.

Notice that, we do not add the $\ell_1$ sparsity penalty until after 50 epochs during the training process. This is to allow the network to freely explore and capture the most salient patterns in the data without prematurely shrinking important weights to zero. This "warm-up" period prevents the model from over-compressing its parameters too early, ensuring it first learns robust representations before enforcing sparsity.

For GREA, which was originally evaluated on polymer datasets, we adopt the authors' default settings. For all other baselines, hyperparameters are automatically optimized using the OPTUNA [1] library.

For baselines, we use the official code package from the authors (DIR, GREA,RPGNN,SSR,DISGEN). For IRM, we implement its graph version based on its official repository. Since source codes of BFGNN[22] is not publically available, we implement it with PYG package. We choose *mean* as the pooling function for baseline methods for its intrinsic generalization, which performs better on larger repeat size compared to Sum. For each task we report *mean±sd* over three random initializations. Each model is trained for up to 400 epochs with early stopping (patience=100), and efficiency was assessed by the epoch at which each model achieved its best performance.

**Repeat-unit Augmentation**  In this paper, GRIN refers to our method with 3-RU augmentation, unless an alternative training scheme is specified in Figure 3. By default, the augmentation training set used for GRIN contains an equal mix of 1-RU and 3-RU samples (1:1 merge ratio), except where otherwise noted in Table 5.

## C Computational Efficiency

We measured training-cost breakdown on the MeltingTemp task using a single NVIDIA A6000 GPU. All models were trained under identical hyperparameter settings and employed the same GCN backbone for a fair comparison. Compared to other baselines, GRIN scales efficiently as the training set doubles. GRIN maintains moderate GPU memory usage (557 MiB) and training time (approximately 764 s). It ranks near the middle of the overall efficiency table, faster than multiple baselines such as SSR and only marginally heavier than its lightweight variant, GRIN-RepAug.

Table 6: Efficiency comparison across models on MeltingTemp task.

| Model | Peak GPU Memory (MiB) | Training Time (s) |
|---|---|---|
| GCN[15] | 455 | 156.64 |
| IRM[3] | 487 | 250.16 |
| RPGCN[21] | 611 | 424.55 |
| GREA[18] | 587 | 652.27 |
| DIR[35] | 533 | 1419.47 |
| SSR[7] | 531 | 1715.97 |
| DISGEN[14] | 629 | 830.42 |
| BFGCN[22] | 565 | 790.26 |
| GRIN-RepAug | 531 | 438.39 |
| GRIN | 557 | 763.58 |

# D Result Details

## D.1 More Results on Effectiveness (Q1)

**More Baselines**  To evaluate the performance compared to attention- and transformer-based models, which provide stronger long-range interaction modeling and global feature mixing capabilities, we conducted additional experiments using GAT [32] and GraphGPS [25] on the MeltingTemp and $O_2$Perm datasets. As shown in Table 7, GRIN outperformed both models.

Tables 8 to 10 presents $R^2$ and RMSE on Test5 and Test10 in addition to Test1 and Test60. We can observe that:

- **Baseline:** All baselines exhibit a steady decline in performance as repeat size increases—from Test1 through Test5 and Test10 to Test60—demonstrating limited size extrapolation.
- **Algorithm Alignment:** Both BFGNN and GRIN-RepAug exhibit a single performance gap between Test1 and Test5, after which their results remain stable through Test10 and Test60. This stability demonstrates the size-generalization benefit conferred by algorithmic alignment. The higher accuracy of GRIN-RepAug compared to BFGNN highlights the additional advantage of alignment with MST.
- **Algorithm Alignment + Augmentation:** GRIN-RepAug's one-time gap between Test1 and other test sets highlights the crucial role of repeat-unit augmentation in bridging the gap between single RU training and multi-repeat generalization. With augmentation, GRIN maintain near-constant $R^2$ and RMSE across Test1, Test5, Test10, and Test60.

These results reinforce the effectiveness of GRIN, which yields robust extrapolation across a wide range of repeat sizes, while conventional methods struggle as chain length grows.

Table 7: Results on homopolymer datasets (MeltingTemp, $O_2$Perm) with 1 repeating unit (Test1) and 60 repeating units (Test60): **GRIN consistently achieves the highest R2 and smallest RMSE.**

| | Model | MeltingTemp | | | | $O_2$Perm | | | |
| --- | --- | --- | --- | --- | --- | --- | --- | --- | --- |
| | | Test1 | | Test60 | | Test1 | | Test60 | |
| | | $R^2\uparrow$ | RMSE$\downarrow$ | $R^2\uparrow$ | RMSE$\downarrow$ | $R^2\uparrow$ | RMSE$\downarrow$ | $R^2\uparrow$ | RMSE$\downarrow$ |
| GIN | GAT [32] | 0.680±0.028 | 63.9±2.8 | 0.579±0.029 | 73.3±2.6 | 0.873±0.066 | 760.4±206.9 | 0.866±0.080 | 774.1±243.0 |
| | GraphGPS [25] | 0.650±0.033 | 66.8±3.2 | 0.563±0.013 | 74.7±1.1 | 0.819±0.168 | 865.2±408.0 | 0.754±0.068 | 1074.1±153.1 |
| | **GRIN-RepAug** | 0.741±0.007 | 57.5±0.8 | 0.707±0.009 | 61.1±0.9 | 0.923±0.008 | 604.7±31.9 | 0.910±0.008 | 655.2±28.6 |
| | **GRIN** | **0.745±0.004** | **57.0±0.4** | **0.746±0.002** | **56.9±0.2** | **0.929±0.002** | **583.4±7.5** | **0.929±0.002** | **581.7±7.2** |

Table 8: Results on homopolymer datasets (GlassTemp, MeltingTemp) with 5 repeating units (Test5) and 10 repeating units (Test10): **GRIN consistently achieves the highest R2 and smallest RMSE.**

| | Model | GlassTemp | | | | MeltingTemp | | | |
| --- | --- | --- | --- | --- | --- | --- | --- | --- | --- |
| | | Test5 | | Test10 | | Test5 | | Test10 | |
| | | $R^2\uparrow$ | RMSE$\downarrow$ | $R^2\uparrow$ | RMSE$\downarrow$ | $R^2\uparrow$ | RMSE$\downarrow$ | $R^2\uparrow$ | RMSE$\downarrow$ |
| GCN | GCN [15] | 0.830±0.005 | 45.8±0.7 | 0.824±0.007 | 46.5±0.9 | 0.670±0.004 | 64.9±0.4 | 0.668±0.004 | 65.1±0.4 |
| | DIR [35] | < 0 | > 100 | < 0 | > 100 | < 0 | > 100 | < 0 | > 100 |
| | GREA [18] | < 0 | > 100 | < 0 | > 100 | < 0 | > 100 | < 0 | > 100 |
| | IRM [3] | < 0 | > 100 | < 0 | > 100 | < 0 | > 100 | < 0 | > 100 |
| | RPGNN [21] | < 0 | > 100 | < 0 | > 100 | < 0 | > 100 | < 0 | > 100 |
| | SSR [7] | < 0 | > 100 | < 0 | > 100 | < 0 | > 100 | < 0 | > 100 |
| | DISGEN [14] | 0.863±0.004 | 41.1±0.6 | 0.855±0.004 | 42.3±0.6 | 0.681±0.007 | 63.9±0.7 | 0.667±0.010 | 65.2±1.0 |
| | BFGCN [22] | 0.855±0.011 | 42.2±1.6 | 0.852±0.013 | 42.6±1.9 | 0.686±0.012 | 63.3±1.2 | 0.684±0.013 | 63.5±1.3 |
| | **GRIN-RepAug** | 0.868±0.010 | 40.3±1.6 | 0.867±0.012 | 40.5±1.8 | 0.712±0.008 | 60.6±0.9 | 0.710±0.008 | 60.9±0.9 |
| | **GRIN** | **0.892±0.001** | **36.5±0.2** | **0.892±0.001** | **36.5±0.2** | **0.746±0.002** | **56.9±0.2** | **0.746±0.002** | **56.9±0.2** |
| GIN | GIN [37] | 0.852±0.001 | 42.9±0.1 | 0.851±0.001 | 42.9±0.1 | 0.673±0.013 | 64.6±1.3 | 0.672±0.014 | 64.7±1.4 |
| | DIR [35] | < 0 | > 100 | < 0 | > 100 | < 0 | > 100 | < 0 | > 100 |
| | GREA [18] | < 0 | > 100 | < 0 | > 100 | < 0 | > 100 | < 0 | > 100 |
| | IRM [3] | < 0 | > 100 | < 0 | > 100 | < 0 | > 100 | < 0 | > 100 |
| | RPGNN [21] | < 0 | > 100 | < 0 | > 100 | < 0 | > 100 | < 0 | > 100 |
| | SSR [7] | < 0 | > 100 | < 0 | > 100 | < 0 | > 100 | < 0 | > 100 |
| | DISGEN [14] | 0.867±0.007 | 40.4±1.1 | 0.860±0.009 | 41.6±1.4 | 0.677±0.018 | 64.2±1.8 | 0.662±0.020 | 65.7±2.0 |
| | BFGIN [22] | 0.853±0.006 | 42.6±0.9 | 0.847±0.008 | 43.3±1.1 | 0.682±0.018 | 63.7±1.8 | 0.680±0.017 | 64.0±1.7 |
| | **GRIN-RepAug** | 0.894±0.003 | 36.2±0.5 | 0.876±0.009 | 39.0±1.5 | 0.736±0.006 | 57.9±0.6 | 0.705±0.001 | 61.4±0.1 |
| | **GRIN** | **0.896±0.001** | **35.7±0.1** | **0.895±0.001** | **36.0±0.1** | **0.740±0.001** | **57.8±0.1** | **0.739±0.002** | **57.7±0.3** |

Table 9: Results on homopolymer datasets (PolyDensity, O$_2$Perm) with 5 repeating units (Test5) and 10 repeating units (Test10): **GRIN consistently achieves the highest R2 and smallest RMSE**.

| | Model | PolyDensity | | | | O$_2$Perm | | | |
|---|---|---|---|---|---|---|---|---|---|
| | | Test5 | | Test10 | | Test5 | | Test10 | |
| | | R$^2$↑ | RMSE↓ | R$^2$↑ | RMSE↓ | R$^2$↑ | RMSE↓ | R$^2$↑ | RMSE↓ |
| GCN | GCN [15] | 0.670±0.026 | 0.125±0.001 | 0.665±0.030 | 0.128±0.006 | 0.876±0.006 | 761.4±17.8 | 0.876±0.006 | 761.5±17.8 |
| | DIR [35] | < 0 | > 1 | < 0 | > 1 | < 0 | > 3000 | < 0 | > 3000 |
| | GREA [18] | < 0 | > 1 | < 0 | > 1 | < 0 | > 3000 | < 0 | > 3000 |
| | IRM [3] | < 0 | > 1 | < 0 | > 1 | < 0 | > 3000 | < 0 | > 3000 |
| | RPGNN [21] | < 0 | > 1 | < 0 | > 1 | < 0 | > 3000 | < 0 | > 3000 |
| | SSR [7] | < 0 | > 1 | < 0 | > 1 | < 0 | > 3000 | < 0 | > 3000 |
| | DISGEN [14] | 0.198±0.105 | 0.198±0.013 | 0.194±0.107 | 0.198±0.014 | < 0 | > 3000 | < 0 | > 3000 |
| | BFGCN [22] | 0.638±0.054 | 0.133±0.010 | 0.635±0.058 | 0.134±0.011 | 0.901±0.014 | 685.1±48.9 | 0.901±0.014 | 685.2±49.1 |
| | **GRIN-RepAug** | 0.720±0.028 | 0.117±0.006 | 0.718±0.030 | 0.118±0.006 | 0.920±0.007 | 621.5±27.9 | 0.915±0.010 | 645.2±28.2 |
| | **GRIN** | **0.745±0.012** | **0.112±0.003** | **0.747±0.011** | **0.112±0.002** | **0.929±0.002** | **581.8±7.5** | **0.929±0.002** | **581.7±7.2** |
| GIN | GIN [37] | 0.645±0.021 | 0.132±0.004 | 0.632±0.028 | 0.134±0.005 | 0.865±0.014 | 801.2±40.7 | 0.865±0.014 | 801.3±40.7 |
| | DIR [35] | < 0 | > 1 | < 0 | > 1 | < 0 | > 3000 | < 0 | > 3000 |
| | GREA [18] | < 0 | > 1 | < 0 | > 1 | < 0 | > 3000 | < 0 | > 3000 |
| | IRM [3] | < 0 | > 1 | < 0 | > 1 | < 0 | > 3000 | < 0 | > 3000 |
| | RPGNN [21] | < 0 | > 1 | < 0 | > 1 | < 0 | > 3000 | < 0 | > 3000 |
| | SSR [7] | < 0 | > 1 | < 0 | > 1 | < 0 | > 3000 | < 0 | > 3000 |
| | DISGEN [14] | 0.154±0.055 | 0.204±0.007 | 0.136±0.057 | 0.206±0.004 | < 0 | > 3000 | < 0 | > 3000 |
| | BFGIN [22] | 0.681±0.016 | 0.125±0.003 | 0.679±0.018 | 0.125±0.004 | 0.913±0.009 | 643.6±32.3 | 0.913±0.009 | 643.6±32.3 |
| | **GRIN-RepAug** | 0.723±0.011 | 0.117±0.002 | 0.721±0.012 | 0.117±0.003 | 0.916±0.003 | 631.1±11.7 | 0.916±0.003 | 631.2±11.7 |
| | **GRIN** | **0.750±0.006** | **0.111±0.001** | **0.751±0.006** | **0.110±0.001** | **0.929±0.006** | **580.9±24.9** | **0.929±0.006** | **580.9±24.9** |

Table 10: Results on copolymer datasets (EA, IP) with 5 repeating units (Test5) and 10 repeating units (Test10): **GRIN consistently achieves the highest R2 and smallest RMSE**.

| | Model | EA | | | | IP | | | |
|---|---|---|---|---|---|---|---|---|---|
| | | Test5 | | Test10 | | Test5 | | Test10 | |
| | | R$^2$↑ | RMSE↓ | R$^2$↑ | RMSE↓ | R$^2$↑ | RMSE↓ | R$^2$↑ | RMSE↓ |
| GCN | GCN [15] | 0.906±0.010 | 0.182±0.009 | ,0.895±0.017 | 0.192±0.016 | 0.909±0.004 | 0.148±0.003 | 0.909±0.004 | 0.149±0.003 |
| | DIR [35] | < 0 | > 1 | < 0 | > 1 | < 0 | > 1 | < 0 | > 1 |
| | GREA [18] | < 0 | > 1 | < 0 | > 1 | < 0 | > 1 | < 0 | > 1 |
| | IRM [3] | < 0 | > 1 | < 0 | > 1 | < 0 | > 1 | < 0 | > 1 |
| | RPGNN [21] | < 0 | > 1 | < 0 | > 1 | < 0 | > 1 | < 0 | > 1 |
| | SSR [7] | < 0 | > 1 | < 0 | > 1 | < 0 | > 1 | < 0 | > 1 |
| | DISGEN [14] | 0.178±0.082 | 0.538±0.027 | 0.179±0.082 | 0.537±0.027 | 0.196±0.056 | 0.442±0.015 | 0.182±0.060 | 0.445±0.016 |
| | BFGCN [22] | 0.883±0.019 | 0.203±0.016 | 0.883±0.018 | 0.203±0.016 | 0.914±0.009 | 0.145±0.007 | 0.913±0.010 | 0.145±0.008 |
| | **GRIN-RepAug** | 0.942±0.006 | 0.143±0.008 | 0.942±0.006 | 0.143±0.008 | 0.937±0.010 | 0.124±0.010 | 0.936±0.010 | 0.124±0.010 |
| | **GRIN** | **0.952±0.002** | **0.129±0.002** | **0.952±0.002** | **0.129±0.002** | **0.949±0.003** | **0.111±0.003** | **0.949±0.003** | **0.111±0.004** |
| GIN | GIN [37] | 0.903±0.006 | 0.185±0.006 | 0.903±0.007 | 0.185±0.006 | 0.915±0.003 | 0.144±0.003 | 0.914±0.004 | 0.144±0.003 |
| | DIR [35] | < 0 | > 1 | < 0 | > 1 | < 0 | > 1 | < 0 | > 1 |
| | GREA [18] | < 0 | > 1 | < 0 | > 1 | < 0 | > 1 | < 0 | > 1 |
| | IRM [3] | < 0 | > 1 | < 0 | > 1 | < 0 | > 1 | < 0 | > 1 |
| | RPGNN [21] | < 0 | > 1 | < 0 | > 1 | < 0 | > 1 | < 0 | > 1 |
| | SSR [7] | < 0 | > 1 | < 0 | > 1 | < 0 | > 1 | < 0 | > 1 |
| | DISGEN [14] | 0.106±0.114 | 0.560±0.037 | 0.102±0.108 | 0.562±0.034 | 0.196±0.056 | 0.442±0.015 | 0.182±0.060 | 0.445±0.016 |
| | BFGIN [22] | 0.918±0.004 | 0.170±0.004 | 0.918±0.004 | 0.170±0.004 | 0.920±0.002 | 0.139±0.002 | 0.919±0.002 | 0.140±0.002 |
| | **GRIN-RepAug** | 0.947±0.004 | 0.137±0.005 | 0.947±0.004 | 0.137±0.005 | 0.954±0.003 | 0.106±0.004 | 0.953±0.003 | 0.107±0.004 |
| | **GRIN** | **0.960±0.001** | **0.119±0.002** | **0.960±0.001** | **0.119±0.002** | **0.962±0.001** | **0.096±0.001** | **0.962±0.001** | **0.096±0.001** |

## D.2 More Results on Repetition-Invariant Representation (Q2)

To further evaluate model generalization beyond limited labeled data, we transferred model check-points trained on the GlassTemp prediction task to 12,764 unlabeled polymers and quantified repeat-size invariance by computing the cosine similarity between embeddings at repeat sizes 1 and 60. As shown in Table 11, GRIN practically learns repetition-invariant representation, with a similarity of 0.999 and zero standard deviation.

Table 11: Results on evaluating representation similarity between 1 repeating unit and 60 repeating units on the large-scale unlabeled polymer dataset: **GRIN achieves 0.999 similarity without standard deviation**.

| Method | Cosine similarity between repeat size 1 and 60 |
|---|---|
| GCN [15] | 0.672±0.119 |
| BFGCN [22] | 0.903±0.060 |
| **GRIN** | **0.999±0.000** |

## D.3 More Results on Ablation Study

**Different Layer-wise Aggregators** To isolate the effect of the max-aggregator in message passing (see Section 3), we use GRIN-RepAug (i.e., without repeat-unit augmentation) to compare three layer-wise aggregation functions—sum, mean, and max—on GlassTemp and MeltingTemp with 1 repeating unit (Test1) and 60 repeating units (Test60). As shown in Table 12, the max-aggregator consistently delivers the best performance on all test sets. Sum outperforms mean on the Test1 for most cases, while it suffers a much larger performance drop at Test60, underscoring that max-aggregation best preserves size-generalization in the MST-aligned architecture.

Table 12: Results on GRIN-RepAug with different layer-wise aggregators: **Max aggregation consistently achieves the highest R2 and smallest RMSE**.

| Encoder | Aggregator | GlassTemp (Test1) | | GlassTemp (Test60) | | MeltingTemp (Test1) | | MeltingTemp (Test60) | |
|---|---|---|---|---|---|---|---|---|---|
| | | $R^2 \uparrow$ | RMSE $\downarrow$ | $R^2 \uparrow$ | RMSE $\downarrow$ | $R^2 \uparrow$ | RMSE $\downarrow$ | $R^2 \uparrow$ | RMSE $\downarrow$ |
| GCN | Max | **0.890±0.001** | **36.8±0.1** | **0.866±0.014** | **40.6±2.0** | **0.741±0.007** | **57.5±0.8** | **0.707±0.009** | **61.1±0.9** |
| | Mean | 0.875±0.005 | 39.3±0.7 | 0.836±0.012 | 44.9±1.6 | 0.694±0.006 | 62.5±0.7 | 0.674±0.002 | 64.5±0.2 |
| | Sum | 0.884±0.002 | 37.7±0.2 | 0.818±0.003 | 48.9±0.4 | 0.707±0.009 | 61.2±0.9 | 0.551±0.055 | 75.7±4.6 |
| GIN | Max | **0.894±0.003** | **36.2±0.5** | **0.876±0.009** | **39.0±1.5** | **0.736±0.006** | **57.9±0.6** | **0.705±0.001** | **61.4±0.1** |
| | Mean | 0.881±0.003 | 38.3±0.5 | 0.838±0.011 | 44.6±1.5 | 0.697±0.007 | 62.2±0.7 | 0.666±0.008 | 65.3±0.8 |
| | Sum | 0.876±0.004 | 39.0±0.6 | 0.815±0.005 | 48.3±0.8 | 0.707±0.007 | 61.2±0.8 | 0.606±0.009 | 70.9±0.8 |

**Repeat-unit Augmentation over Baselines** To assess the effect of augmentation, we apply RepAug with 3 repeating units to the GCN and GIN baselines and test on GlassTemp and MeltingTemp with 1 repeating unit (Test1) and 60 repeating units (Test60). The results summarized in Tables 13 and 14 show that RepAug yields consistent improvements: on Test1, both GCN+RepAug and GIN+RepAug achieve higher $R^2$ and lower RMSE compared to their non-augmented counterparts. The improvements are more pronounced on Test10, demonstrating that repeat-unit augmentation significantly enhances size-extrapolation performance not only on our GRIN architecture. Figure 2 (a) visualizes the polymer latent representations from GCN and GCN+RepAug with t-SNE.

Table 13: Results on GCN baseline with 3-RU augmentation: **RepAug improves performance on both 1 repeating unit (Test1) and 10 repeating units (Test10) for GlassTemp and MeltingTemp.**

| Task | Model | Test1 | | Test10 | |
|---|---|---|---|---|---|
| | | $R^2 \uparrow$ | RMSE $\downarrow$ | $R^2 \uparrow$ | RMSE $\downarrow$ |
| GlassTemp | GCN[15] | 0.878±0.001 | 38.7±0.1 | 0.824±0.007 | 46.5±0.9 |
| | GCN+RepAug | **0.884±0.004** | **37.8±0.6** | **0.872±0.004** | **39.8±0.5** |
| MeltingTemp | GCN[15] | 0.693±0.002 | 62.6±0.2 | 0.668±0.004 | 65.1±0.4 |
| | GCN+RepAug | **0.694±0.003** | **62.5±0.3** | **0.683±0.002** | **63.6±0.2** |

Table 14: Results on GIN baseline with 3-RU augmentation: **RepAug improves performance on both 1 repeating unit (Test1) and 10 repeating units (Test10) for GlassTemp and MeltingTemp.**

| Task | Model | Test1 | | Test10 | |
|---|---|---|---|---|---|
| | | $R^2 \uparrow$ | RMSE $\downarrow$ | $R^2 \uparrow$ | RMSE $\downarrow$ |
| GlassTemp | GIN[37] | 0.882±0.003 | 38.1±0.5 | 0.851±0.001 | 42.9±0.1 |
| | GIN+RepAug | **0.887±0.004** | **37.2±0.6** | **0.882±0.003** | **38.0±0.5** |
| MeltingTemp | GIN[37] | 0.697±0.007 | 62.2±0.7 | 0.672±0.014 | 64.7±1.4 |
| | GIN+RepAug | **0.701±0.009** | **61.8±0.9** | **0.694±0.006** | **62.5±0.7** |

# E   Future Work

**Limitations**   The scale of labeled datasets used in our experiments remains limited, as collecting large, high-quality polymer datasets is inherently challenging. For example, researchers have spent nearly 70 years compiling only about 600 polymers with experimentally measured oxygen permeability in the Polymer Gas Separation Membrane Database [29]. We only partially address the problem by evaluating model generalization on a larger unlabeled polymer dataset by computing representation similarity, as described in appendix D. Another limitation is that incorporating augmented training samples in GRIN leads to a modest increase in training time and GPU memory consumption, as reported in appendix C.

While GRIN already establishes a strong foundation for learning repetition-invariant polymer representation, several promising directions could further reinforce its capabilities:

- **Advanced Algorithm Alignment.** Building on our MST-aligned method, future work could explore alignment with more sophisticated dynamic-programming algorithms, which may enable GRIN to learn more comprehensive intra-monomer connections especially for branched or cross-linked polymer architectures.

- **Broader Material Domains.** Our current copolymer dataset comprises two primary architectures: block copolymers, with large contiguous runs of the same monomer, and alternating copolymers, featuring `-A-B-` repetition at first, each generating distinct structural motifs that affect material properties. Future work could incorporate random copolymers, whose monomer order lacks a fixed pattern, and apply the GRIN framework to each architecture, uncovering architecture-specific invariance and informing tailored representation strategies.

