# OpenReview forum: "Learning Repetition-Invariant Representations for Polymer Informatics"
_NeurIPS.cc/2025/Conference — NeurIPS 2025 poster_

### Official Review · Reviewer_W2de · 2025-06-27

**Clarity:** 2
**Significance:** 3
**Originality:** 3
**Rating:** 4
**Confidence:** 3

**Summary:**

This paper addresses the challenge that conventional graph neural networks, which model polymers using only a single repeating unit, produce embeddings that drift as the number of monomer repeats varies. To overcome this, the authors propose Graph Repetition Invariance, which augments polymer graphs with multiple repeat units and aligns GNN message passing to a maximum spanning tree via max‐aggregation and a sparsity penalty—ensuring that only the most informative edges drive representation updates. Three‐unit augmentations suffice for learning truly repeat‐invariant embeddings and demonstrate, across both homopolymer and copolymer benchmarks, that GRIN yields stable, size‐agnostic feature vectors that outperform existing baselines in property prediction tasks

**Questions:**

I don’t quite understand the example from lines 27–33. Could you provide a concrete illustration? Moreover, do large language models exhibit this problem? If not, perhaps we can handle such invariance naturally via augmentations present in real data.

What is the meaning of Figure 2? How should we interpret the right-hand panel?

I’d like to confirm the overall pipeline: we start with a set of polymers, extract their monomer units, generate augmented data from those monomers, append it to the training set, and then train a GRIN model. Is that correct?

Beyond these geometric deep learning baselines, what about kernel-based methods? I believe repeating patterns could be addressed using domain knowledge—perhaps via motif-inspired embeddings. Could we design (deep) graph kernels to tackle this problem?

How would models like Residual Gated Graph ConvNets, which have an inherent sequential design, perform on this task?

The improvement of the GRIN variant over the baseline—and of training on G₁+G₃ versus G₁ alone—is not very pronounced given the limited training data. Could you explain this? Does it imply that encoding such inductive biases as hard constraints is unnecessary when a large amount of training data is available?

**Ethical Concerns:**

["NO or VERY MINOR ethics concerns only"]

**Final Justification:**

I support the acceptance

**Quality:**

3

**Strengths And Weaknesses:**

Strengths:
1. The paper is generally well written.
2. The experimental section is comprehensive
3. It poses an interesting, timely question in the graph-learning domain that could inspire future research. Despite minor issues, the work has the potential to benefit the broader community.

Weakness:
1. I think there are some points not clearly stated. I will check with them in the question part.
2. The theoretical part is more akin to intuition than rigorous evidence.
3. The scale of datasets seems limited

---

> ### Author Rebuttal · Authors · 2025-07-31
>
> ## W1 & Q1: Example from lines 27-33
> > I don’t quite understand the example from lines 27–33. Could you provide a concrete illustration?
>
> For L27-33, we draw an analogy between repeated phrases in language and repeated monomer units in polymers. For instance, in sentiment analysis, phrases like “very fair”, “very very very fair”, "very fair very fair" typically express the same sentiment. The sentiment model should recognize that the core meaning remains unchanged despite repeated modifiers.
>
> Similarly, a polymer with one repeating unit (1RU) and many RUs should be assigned highly similar feature representations since their chemical identity remains unchanged.
>
> > Moreover, do large language models exhibit this problem? If not, perhaps we can handle such invariance naturally via augmentations present in real data.
>
> ### Real-world augmentations benefit language models but transfer poorly to graph domains.
> * **In NLP, repetition is ubiquitous and cross-domain transferable.**
> Natural language corpora consistently exhibit repeated structures across domains (e.g., “very very good”) such as news, literature, and scientific writing [3]. Allowing LLMs to learn repeat-invariant patterns through large-scale pretraining without task-specific augmentation.
> * **In graph domains, repetition patterns are highly domain-specific.**
> As explicitly noted by Chen et al., 2022 [4], repetition in polymer graphs (see Figure 1) is highly domain-specific, distinguishing them from small molecules and social graphs.
> * **Polymer data scarcity**
> As we discussed in W3, labeled polymer data is limited due to the high cost and complexity of obtaining reliable property labels.
>
> ## Q2: Figure 2
> >What is the meaning of Figure 2?
>
> Figure 2 visualizes how repeat invariance improves progressively—from GCN’s repeat-size-based clustering, to GRIN’s repeat-invariant latent space.
>
> It shows a t‑SNE projection of polymer embeddings on the GlassTemp task for four models. We visualize 50 polymers, each represented by 20 embeddings (repeat sizes 1–20), colored ight (1RU) to dark (20RU).
>
> * **GCN (leftmost):** Embeddings group by repeat size (e.g., all 20RU points, darker one, cluster together), not polymer identity, indicating poor invariance (sim ~ 0.73).
> * **GCN+RepAug:** Light-to-dark stripes emerge—indicating that repeat variants of the same polymer start grouping together, as RepAug promotes consistency across repeat sizes (sim ~ 0.85).
> * **GRIN-RepAug and GRIN (right panels):** Each polymer’s 20 repeat variants collapse into a tight, size-invariant cluster (sim > 0.98).
>
> This trend is also quantified:
>
> | Model       | Cosine similarity between repeat size 1 and 60 |
> | ----------- | ------------------------------------ |
> | GCN         | 0.731±0.154  |
> | GCN+RepAug  | 0.849±0.014  |
> | GRIN-RepAug | 0.980±0.000  |
> | GRIN        | **0.998±0.000**|
>
> To make it clearer, we revised the figure caption as:
>
> - "Figure 2: t‑SNE visualization of polymer embeddings from GCN and GRIN (ours) across repeat sizes (1–20 RUs) on the glass transition task.
> Points are colored by repeat count (light = 1RU, dark = 20RU). (a) GCN produces inconsistent embeddings for different repeat sizes of the same polymer, clustering by size (same color) rather than identity; our augmentation introduces light-to-dark stripes, indicating improved alignment of repeat variants. (b) GRIN learns repeat‑invariant representations: each polymer’s variants form tight, size‑independent clusters, further improved with augmentation."
>
> >How should we interpret the right-hand panel?
>
> The last two panels show GRIN-RepAug and GRIN, where each polymer’s 20 variants (across repeat sizes) form tight, color-mixed clusters. This indicates that the model outputs nearly identical embeddings for different repeat sizes.
>
> ## Q3: Overall pipeline
> > I’d like to confirm the overall pipeline:...
>
> The reviewer’s understanding of the pipeline is generally correct for model training. For inference, the trained model can take polymers with arbitrary repeat lengths and produce invariant representations.
>
> ## Q4: Kernel-based methods
> >Beyond these geometric deep learning baselines, what about kernel-based methods? I believe repeating patterns could be addressed using domain knowledge—perhaps via motif-inspired embeddings.
>
> We agree that domain-specific methods like graph kernels or motif-counting approaches are promising directions for polymer representation, especially since repeating patterns are core to polymer structure.
> > Could we design (deep) graph kernels to tackle this problem?
>
> However, designing a polymer-aware kernel or motif library is non-trivial. The key structural features often span both intra- and inter-monomer interactions, are sensitive to anchor positions within the chain, and vary with copolymer patterns (e.g., block, alternating, or random). Additionally, to support size invariance, such motifs must be periodic and invariant to repeat count—a challenge for traditional motif extraction.
>
> In this work, we focused on GNN-based methods to learn such patterns automatically, without requiring an explicit library of motifs.
>
> ## Q5: Residual Gated Graph ConvNets
> >How would models like Residual Gated Graph ConvNets, which have an inherent sequential design, perform on this task?
>
> ### GRIN outperformed ResGatedGCN
> We conducted new experiments on ResGatedGCN [5] and a pure sequence baseline, the SMILES-based LSTM [6]. GRIN outperformed ResGatedGCN [5] as follows:
>
> #### MeltingTemp
>
> | Model            | Test1 R²↑   | Test1 RMSE↓ | Test60 R²↑  | Test60 RMSE↓ |
> | ---------------- | ----------- | ----------- | ----------- | ------------ |
> | ResGatedGCN      | 0.710±0.003 | 60.8±0.3    | 0.454±0.048 | 83.5±3.7     |
> | LSTM             | 0.610±0.032 | 70.6±2.8    | <0          | >3000        |
> | GRIN-RepAug(GCN) | 0.741±0.007 | 57.5±0.8    | 0.707±0.009 | 61.1±0.9     |
> | GRIN(GCN)        | 0.745±0.004 | 57.0±0.4    | 0.746±0.002 | 56.9±0.2     |
>
> ## Q6: The improvement of GRIN
>
> >The improvement of the GRIN variant over the baseline—and of training on G₁+G₃ versus G₁ alone—is not very pronounced given the limited training data. Could you explain this?
>
> Across all tasks, GRIN consistently ranks among the top and remains stable, while baseline models often fluctuate—some peak on specific datasets and degrade significantly at larger repeat sizes.
> * Testing on a single repeating unit (Test1), GRIN consistently outperforms BFGNN across homopolymer tasks such as MeltingTemp (\~5%) and density (\~6–7%), as well as more challenging copolymer tasks such as EA (\~5%) and IP (\~5%).
> * Testing on 60 repeating units (Test60), GRIN’s advantages become more pronounced as it maintains strong performance across all tasks. Five out of eight baselines have negative $R^2$, contrasting GRIN’s generalization to different repeating sizes.
>
> GRIN-RepAug is an ablated variant of GRIN without repetition augmentation. Its strong performance demonstrates the effectiveness of the new neural architecture under limited training data, but it does not imply that the inductive bias becomes unnecessary with larger datasets. From Tables 3, 4, 10 and 11, we found the new architecture remains important even when data augmentation is applied.
>
> >Does it imply that encoding such inductive biases as hard constraints is unnecessary when a large amount of training data is available?
>
> As discussed in W3 and Q1, polymer datasets are limited and costly to label. GRIN addresses this by embedding repetition invariance as an architectural prior, enabling generalization under realistic, small-data regimes.
>
> ## W2: Theory part
> >The theoretical part is more akin to intuition than rigorous evidence.
>
> The theoretical analysis in Section 3.3 provides an algorithmic insight aiming to explain how specific architectural choices—such as max aggregation and L1 sparsity—promote consistent reasoning across varying graph sizes.
>
> Discussion at L189–L204 show that, under these design choices and augmentation schemes, the model produces stable outputs regardless of repeat length. This aligns with our goal: to show that GRIN encourages repeat-invariant behavior.
>
> ## W3: Limited dataset scale
>
> ### GRIN generalizes better on larger unlabeled datasets for repeat-size invariance.
>
> Collecting large-scale labeled polymer datasets is challenging. For example, researchers have spent nearly 70 years compiling only about 600 polymers with experimentally measured oxygen permeability in the Polymer Gas Separation Membrane Database [1].
>
> To address your concern, we used a larger unlabeled polymer dataset from [2] to evaluate the effectiveness of our method. We applied model checkpoints from the GlassTemp task to ~13k unlabeled polymers and measured repeat-size invariance using cosine similarity between embeddings at repeat sizes 1 and 60. The results below indicate that GRIN generalizes better to large repeat sizes than other GNNs on this large-scale dataset.
>
> | Method       | Cosine similarity between repeat size 1 and 60 |
> | ----------- | --------------- |
> | GCN         | 0.672± 0.119    |
> | BFGNN       | 0.903± 0.060    |
> | GRIN        | **0.999±0.000** |
>
> ## References
>
> [1] A Thornton, L Robeson, B Freeman, and D Uhlmann. 2012. Polymer Gas Separation Membrane Database.
>
> [2] Otsuka S, Kuwajima I, Hosoya J, Xu Y, Yamazaki M. PoLyInfo: Polymer database for polymeric materials design. International Conference on Emerging Intelligent Data and Web Technologies, 2011.
>
> [3] Don't Stop Pretraining: Adapt Language Models to Domains and Tasks. (ACL 2020)
>
> [4] Graph Representation Learning for Polymer Property Prediction. (NeurIPS 2022)
>
> [5] Residual Gated Graph ConvNets.
> [6] Chen, G., Tao, L., & Li, Y. (2021). Predicting Polymers' Glass Transition Temperature by a Chemical Language Processing Model. Polymers, 13(11), 1898.

---

> > ### Comment · Reviewer_W2de · 2025-08-03
> >
> > Thanks for the detailed response, which has well addressed my concerns. I support the acceptance of this paper.

---

> > > ### Author Response · Authors · 2025-08-04
> > >
> > > We are encouraged that your concerns have been addressed. Thank you again for your valuable and helpful feedback! We welcome any new discussion.

---

### Official Review · Reviewer_vHcv · 2025-06-28

**Clarity:** 3
**Significance:** 3
**Originality:** 2
**Rating:** 4
**Confidence:** 4

**Summary:**

The paper considers the problem of applying GNNs to polymers, which are composed of repeated structures. They propose a method that (1) encodes polymers as a graph, (2) uses max aggregation GNNs, (3) augments the training data with polymers of different numbers of repeating units, and (4) encourages sparsity with L_1 regularisation. They provide some theoretical results about the correspondence of their model to the MST algorithm, and about how many repeated units need to be added during data augmentation. They perform experiments showing model performance across 6 datasets against a number of baselines, and also perform an ablation study on the repeat size and proportion of augmented graphs.

**Questions:**

1. L51 "The MST criterion further preserves the most informative backbone of the polymer graph." -> what does this mean? It is likewise unclear to me what "dynamic programming(DP)-style updates enforce a fixed reasoning procedure" means.
2. L78 What do you mean by "causal" information?
3. L120 - L121 "With residual connection h, any node not chosen by max-aggregation simply skips the update and retains its previous embedding unchanged." -> What does this mean? And how is it reflected in the above equation?
4. L149 -> I can see a correspondence between MST and max aggregation, but where here is the "theoretical guarantee" that was spoken of?
5. L257 "MST-aligned max aggregation" -> is it not just max aggregation, which itself is "MST-aligned"? This phrasing is a bit confusing to me.

Minor comments:
1. Figure 2 is not explained well in the caption. Also, the figure doesn't show explicitly which polymers belong to the same class (i.e. are variants of one another with a different number of repeating units)
2. L41 -> could be nice to quantify this, rather than just motivating visually
3. L104 -> the graph encoding would benefit from a visualisation
4. L127 "sparsity constraint" -> this is the first time the sparsity constraint is mentioned. It is only defined later
5. L132 -> typo

**Ethical Concerns:**

["NO or VERY MINOR ethics concerns only"]

**Final Justification:**

I have increase my score to a borderline accept. The authors did well to address my concerns in rebuttal.
However, the paper does need a lot of changes, particularly the ones that the the authors agreed to in my review and over the course of our conversation: the paper messaging around its contribution (and many other areas) are not clear in the version submitted to the conference.

If these changes are addressed, I think the paper should be accepted: the work is a nice application, has strong results, and uses some neat theoretical insights. However, given the number and extent of the required changes, I cannot increase my score beyond a borderline accept, and I somewhat lean in the direction of recommending that the authors substantially edit the paper and then resubmit.

**Limitations:**

Limitations of the work are not discussed at all. Future work is included in the appendix.

The authors need to seriously consider their paper and the NeurIPS requirements to discuss limitations, as stated in the checklist item 2.
The bullet points under the guidelines provide a thorough explanation of many things the authors could have included in their paper as limitations. As one simple example: the paper should have considered the limitation of using soft invariance vs guaranteed invariance.

**Quality:**

1

**Strengths And Weaknesses:**

Strengths:
1. Strong model performance - outperformed the baselines in thorough experiments.
2. Nice motivation for the solution: invariance to the number of repeating units is a very fitting invariance to encode for the problem being solved.
3. Generally technically sound - terms were mostly well defined and explained.
4. Writing was clear, concise, and readable.

Weaknesses:
1. The claims of the paper are often overstated and do not reflect the actual contribution. This is the primary reason I have assigned a score of 1 for Quality and my biggest reason to reject the paper. My current perspective is that large portions of the paper need to be re-written with the true contributions in mind. To expand on this:

1.1. The paper constantly refers to their embeddings as "repeat-invariant". However, nothing about their model guarantees this, and the embeddings shown in Figure 2 show that GRIN is not actually invariant, just approximately so. E.g. on L252, the phrase "demonstrating true repetition invariance" is used. By contrast, GCNs are truly invariant to node permutation, and other GNNs are also invariant to operators such as translation and rotation. In the paper, there needs to be a much clearer distinction between true invariance and approximate invariance.

1.2 The paper claims to "align the model with the MST construction process" - this is very vague terminology and the actual theory presented in section 3.3.1 does not correspond to the language used. Furthermore, the actual "alignment" is between any GNN using max aggregation and a sparsity constraint, so is not particular to the GRIN model. It is also not clear why the "alignment" with MST is actually useful to the problem of repetition invariance - this needs to be expanded on.

1.3 The paper phrases their contribution as a new model (GRIN). However, this can be separated into (1) some common choices for GNN architecture (max aggregation and sparsity constraints), (2) a graph encoding scheme, and (3) data augmentation. (2) and (3) seem to be the main novelty of the paper to me, so the messaging should be centred around them, rather than pitching it as a new model.

2. Some of the theory could be made clearer, and the impact of the results more clearly stated up-front. For example, section 3.3.2 was dense and hard to understand - it should be expanded, and the explanation given on L174 of why the theory is useful explained better up-front, so the motivation is clear. At other times, terms are not defined or cited (e.g. "L-Lipschitz" on L181).

3. Some motivation is lacking, and could be better shown in the experiments. For example, on L99, it is stated that "sum grows unbounded" as motivation for using max aggregation. However, it is not clear why why is this a problem. In practice, the repeat size will not become infinitely large. This argument also applies to any graph as the size increases. This claim could be demonstrated empirically by using sum aggregation vs max in the GRIN framework and showing the benefits of max.

4. Some of the experimental results are underwhelming. For example, the performance gain of GRIN over other models such as BFGIN seems quite marginal. This seems to undercut the thesis of the necessity of repetition invariance.

---

> ### Author Rebuttal · Authors · 2025-07-31
>
> ## W1.1: Invariance
> >The paper constantly refers to their embeddings as "repeat-invariant". ...
>
> #### GRIN achieves a cosine similarity of 0.999 ± 0.000 between repeat sizes 1 and 60 on ~13K unlabeled polymer SMILES.
> Thank you for pointing out the distinction between exact mathematical invariance and the practical invariance we observe. We appreciate this suggestion and have revised the phrasing to ensure greater rigor. Our quantitative analysis shows that the embedding similarity between repeat sizes 1 and 60 is very close to 1, indicating strong repeat invariance.
>
> Specifically, we have updated the manuscript as follows:
> * **L252:** “…demonstrating true repetition invariance.” → to: “…demonstrating repetition invariance.”
> * We re-scanned the manuscript and found no other instances of absolute phrases such as “true repetition invariance.” To avoid ambiguity, we will add the following clarifying sentence in the Introduction:
>   *“Throughout this work, we use ‘repetition-invariant’ to denote practical invariance, defined as a latent representation similarity above 0.99 across repeat sizes.”*
>
> To quantify repetition invariance, we evaluated the model checkpoints from the GlassTemp task on ~13K unlabeled polymer SMILES [1]. We computed the cosine similarity between embeddings at repeat sizes 1 and 60. As shown below, GRIN achieved a cosine similarity of 0.999 ± 0.000.
> | Method       | Cosine similarity between repeat size 1 and 60 |
> | ----------- | --------------- |
> | GCN         | 0.672± 0.119    |
> | BFGNN       | 0.903± 0.060    |
> | GRIN        | **0.999±0.000** |
>
> >and the embeddings shown in Figure 2 show that GRIN is not actually invariant, just approximately so.
>
> We calculated the mean cosine similarity between embeddings at repeat sizes 1 and 60 on the GlassTemp test set. GRIN achieves 0.998 (± 0.000).  (Full table please refer to Minor 2)
>
> ## W1.2: MST alignment
> >The paper claims to "align the model with the MST construction process" - very vague terminology
>
> >  "dynamic programming(DP)-style updates enforce a fixed reasoning procedure" means.
>
> We follow the term "neural algorithmic alignment" [4], aiming to design neural architectures that align structurally with specific DP paradigms.
> * **Algorithmic paradigm:** MST algorithms build a tree by greedily selecting sparse, non-redundant edges.
> * **Neural architecture:** GRIN’s Max aggregation selects the strongest incoming message, while the sparsity penalty suppresses less informative edges. This encourages sparse decision paths and avoids uniform mixing of neighbor information, resulting in a tree-like information flow qualitatively aligned with MST-style connectivity.
>
> GRIN's update scheme mimics DP-style reasoning: a fixed local rule is applied layer-wise, resulting in deterministic update paths. It avoids arbitrary neighbor mixing, as seen in mean or soft attention aggregation.
>
> We clarified the phrasing as:
> * **L49:** "GRIN aligns message passing with the edge-greedy logic" → to "GRIN draws inspiration from the edge-greedy logic"
> * **L53-54:** "aligns it with the MST construction process." → to "heuristically encourages alignment with ..."
>
> >the actual "alignment" is not particular to the GRIN model.
>
> Most GNNs adopt sum or mean aggregation [2,3]. The combination of max aggregation with an L1 sparsity penalty is underexplored and, to our knowledge, has not been theoretically or empirically studied for polymer property prediction. This architectural choice, together with repeat-unit augmentation, distinguishes GRIN from prior work.
>
> >  why the "alignment" with MST is actually useful
>
> >L51 "The MST criterion further preserves the most informative backbone of the polymer graph."
>
> The inductive bias of MST is particularly beneficial for repetition invariance. When a polymer graph is expanded with repeated units, many edges are redundant. MST-style selection prevents uniform mixing of the duplicates and concentrates information flow along maximal, consistent paths that capture the core structural features independent of graph size.
>
> ## W1.3 GRIN phrases
> Thank you for the suggestion. The main text reflects this point in Lines 47–49: “In this work, we propose Graph Repetition INvariance (GRIN), a method that combines algorithm-alignment with repeat-unit augmentation to address the challenge of repetition-invariant representation learning.”
>
> ## W2: Clarity of theory
> Thanks for your suggestion. We revised and moved **L174:** "With repeat-augmented polymers $P_m$ ($m$ ≥ 2), the model can apply same update rule at each layer, yielding a constant output $y^⋆$, irrespective of the repeat size." to the beginning of Section 3.3.2 (after L153).
>
> We revised the sentence on **L181:** "...is $L$-Lipschitz with $L$<1..."
> → to "... is assumed to be $L$-Lipschitz, i.e., it satisfies $|f(x)−f(x′)|≤ L|x−x′|$ for all inputs $x$, $x′$ and some constant $L$<1...""
>
> ## W3: Motivation for max aggregation
>
> In Appendix C.2 (Lines 417–424, Table 9), we already evaluated different aggregation schemes within the GRIN-RepAug (GCN/GIN backbone). Both sum and mean aggregators lead to varying degrees of performance degradation.
>
> ## W4: Experimental Results
>
> ### GRIN consistently outperforms baselines across repeat sizes, with especially strong generalization on Test60 where many baselines fail.
>
> Testing on single repeating units (Test1), GRIN consistently outperforms BFGNN across homopolymer tasks such as MeltingTemp (\~5%) and density (\~6%), and more challenging copolymer tasks EA (\~5%) and IP (\~5%).
>
> On Test60, GRIN’s advantages become more pronounced as it maintains strong performance across all tasks. While 5/8 baselines have negative $R^2$.
>
> ## Q2: L78
>
> “Causal information” refers to task-relevant structural features that remain consistent regardless of the repeat count. We acknowledge that “causal” may suggest a formal treatment (e.g., counterfactuals) which we do not pursue here. To avoid confusion, we revised **L78:** "causal information” → to “key structural information”
>
> ## Q3: L120 - L121
>
> Thank you for catching this. In our design, skip arises implicitly: max aggregation selects one dominant neighbor per dimension, and L1 sparsity suppresses the rest, making their contributions negligible—functionally approximating an identity mapping for non-winning paths.
> We revised **L121:** "With residual connection h, any node not chosen by max-aggregation simply skips the update..." → to "With previous state $h^{l-1}$, any non‑max neighbour contributes zero to $U^l$ leaves corresponding feature dimensions remain unchanged..."
>
> ## Q4: L149
> The guarantees are stated in Section 3.3:
> * Proposition 3.2: Training on {$P_1$, $P_n$} with L1 sparsity yields predictions invariant to chain length ($m ≥ 2$) under the hyperchain abstraction, as all node types are exposed and non-dominant paths are suppressed.
> * Proposition 3.3: $n = 3$ is the minimal augmentation introducing a degree‑2 node and enabling true two-branch competition; larger $n$ yields diminishing returns under the $L$‑Lipschitz contraction, consistent with observed performance saturation.
>
> ## Q5: L257
> We intended to emphasize that the max-aggregation was inspired by MST, but saying “MST-aligned max aggregation” can be redundant. We simplified this to avoid confusion:
> * **L257:** “... MST-aligned max aggregation ...”
> → to “... MST-aligned aggregation ...”
>
> ## Minor 1: Figure 2
> > Figure 2 is not explained well in the caption.
>
> The caption is revised as: "Figure 2: ... glass transition task. Points are colored by repeat count (light = 1RU, dark = 20RU). (a) GCN produces inconsistent embeddings for different repeat sizes of the same polymer, clustering by size (same color) rather than identity; our augmentation introduces light-to-dark stripes, indicating improved alignment of repeat variants. (b) ..."
> > Also, the figure doesn't show explicitly which polymers belong to the same class...
>
> Though Figure 2 does not label polymer identities, it clearly visualizes improved repeat-size invariance across models—supported by Minor 2’s scores:
> - **GCN**: Embeddings cluster by repeat count—e.g., all dark points (20RU) group together, showing low invariance (sim ~ 0.73).
> - **GCN + RepAug**:  Light-to-dark stripes emerge, showing variants of same polymers begin to align, shifting clustering from repeat size to identity (sim ~ 0.85).
> - **GRIN w/wo RepAug**: Each polymer’s embeddings form tight, size-independent clusters, indicating strong invariance (sim > 0.98).
>
> ## Minor 2: L41
> Thanks for your comment. We computed embedding similarity between repeat sizes 1 and 60 on GlassTemp and found that RepAug improves GCN stability, while GRIN variants achieves consistency.
> | Model       | Cosine similarity between repeat size 1 and 60 |
> | ----------- | ------------------------------------ |
> | GCN         | 0.731±0.154  |
> | GCN+RepAug  | 0.849±0.014  |
> | GRIN-RepAug | 0.980±0.000  |
> | GRIN        | **0.998±0.000**|
>
> ## Minor 3: L104
> Thanks for your comment. We added a visualization for this part. According to the conference's protocol, we are not allowed to attach figure or link here.
>
> ## Minor 4: L127
> Thanks for your comment, we defined it at first mention: **Revised (L124–L131):** “Max aggregation imposes a selection bias toward the strongest neighbor, we add an L1 sparsity penalty on the message and update networks to suppress non‑dominant pathways (Eq. 3), encouraging a sparse, MST‑like backbone.”
>
> ## Minor 5: L132
> We corrected “constrcuted” → “constructed” in the description.
>
> ## References
>
> [1] PoLyInfo: Polymer database for polymeric materials design. International Conference on Emerging Intelligent Data and Web Technologies,  2011.
>
> [2] How Powerful Are Graph Neural Networks? (ICLR 2019)
>
> [3] Physical Pooling Functions in Graph Neural Networks for Molecular Property Prediction, 2022.
>
> [4] Graph neural networks extrapolate out-of-distribution for shortest paths, 2025.

---

> > ### Comment · Reviewer_vHcv · 2025-08-01
> > **Response**
> >
> > Thank you to the authors for a thorough response, and for addressing most of my concerns.
> > Here follows a few remaining concerns / disagreements on my side:
> >
> > > Most GNNs adopt sum or mean aggregation [2,3]. The combination of max aggregation with an L1 sparsity penalty is underexplored and, to our knowledge, has not been theoretically or empirically studied for polymer property prediction. This architectural choice, together with repeat-unit augmentation, distinguishes GRIN from prior work.
> >
> > Whilst sum or mean is the common choice for GNNs, max is still widely used. L1 sparsity is also a very common model penalty.
> > It may be the case that the particular combination has not been theoretically or empirically studied for polymer property prediction, but I think that both architectural choices are so common that simply combining them cannot be pitched as a new architecture, in the messaging of the paper. I think it would be far clearer to state something like "we choose max aggregation and L1 sparsity, and now here are some theoretical and empirical reasons why this is a good idea".
> >
> > > The inductive bias of MST is particularly beneficial for repetition invariance. When a polymer graph is expanded with repeated units, many edges are redundant. MST-style selection prevents uniform mixing of the duplicates and concentrates information flow along maximal, consistent paths that capture the core structural features independent of graph size.
> >
> > One of my concerns here is that "many edges are redundant" is not the same as "only one edge is relevant" in each message passing step, which is what max aggregation yields (for each vector element). Furthermore, I do not find it helpful to frame this in terms of "MST-style selection", which somewhat obfuscates the fact that it's ultimately max aggregation which concentrates information flow along single paths.
> >
> > In Appendix C.2 (Lines 417–424, Table 9), we already evaluated different aggregation schemes within the GRIN-RepAug (GCN/GIN backbone). Both sum and mean aggregators lead to varying degrees of performance degradation.
> >
> > > Thanks for referring me to this. I would suggest moving this into the main section of the paper, as it is good empirical motivation for your architectural choices (or at least mentioning these results in the main section of the paper).

---

> > > ### Author Response · Authors · 2025-08-01
> > >
> > > ## 1
> > > > Whilst sum or mean is the common choice for GNNs, max is still widely used. L1 sparsity is also a very common model penalty. It may be the case that the particular combination has not been theoretically or empirically studied for polymer property prediction, but I think that both architectural choices are so common that simply combining them cannot be pitched as a new architecture, in the messaging of the paper. I think it would be far clearer to state something like "we choose max aggregation and L1 sparsity, and now here are some theoretical and empirical reasons why this is a good idea".
> > >
> > > Thanks for your suggestion. We agree that these components are indeed well-known. We carefully reviewed the manuscript and confirm that we do not describe the combination of max aggregation and L1 sparsity as a new architecture.
> > >
> > > Throughout the paper, we only refer to GRIN as the novel method and do not attribute novelty to the architecture choice in isolation as:
> > >
> > > * **Lines 6–8**: “To address this challenge, we introduce Graph Repetition Invariance (GRIN), a novel method to learn polymer representations that are invariant to the number of repeating units…”
> > > * **Line 47**: “In this work, we propose Graph Repetition INvariance (GRIN), a method that combines…”
> > > * **Lines 91–93**: “We present Graph Repetition Invariance (GRIN), a novel framework that (i) augments polymer graphs…”
> > > * **Line 256**: “We made the first attempt to achieve repetition-invariant representation learning…”
> > >
> > > ## 2
> > >
> > > > One of my concerns here is that "many edges are redundant" is not the same as "only one edge is relevant" in each message passing step, which is what max aggregation yields (for each vector element).
> > >
> > > Thank you for the thoughtful feedback. We acknowledge that “many edges are redundant” is not equivalent to “only one edge is relevant”. Our earlier reply—“When a polymer graph is expanded with repeated units, many edges are redundant”—was intended to motivate why MST-inspired reasoning may help with repetition invariance, by preventing uniform mixing and promoting consistent information flow across duplicated substructures. Answering:
> > > >'why the "alignment" with MST is actually useful'
> > >
> > > While the two notions are not strictly identical, in the context of large polymer graphs (e.g., with 60 repeating units), max aggregation—perhaps at the cost of discarding some moderately relevant signals—effectively prunes less informative paths. This selective propagation plays a key role in achieving approximate invariance across different graph sizes.
> > >
> > > > Furthermore, I do not find it helpful to frame this in terms of "MST-style selection", which somewhat obfuscates the fact that it's ultimately max aggregation which concentrates information flow along single paths.
> > >
> > > Thank you for pointing this out. As our revision in response to W1.2, we will make sure to clarify that the core mechanism is max aggregation combined with sparsity, which is heuristically aligned with MST principles.
> > >
> > >
> > > ## 3
> > > > Thanks for referring me to this. I would suggest moving this into the main section of the paper, as it is good empirical motivation for your architectural choices (or at least mentioning these results in the main section of the paper).
> > >
> > > Thank you for the suggestion. We will move this empirical motivation into Section 3.3.1, before Line 147 to better support our architectural choices.
> > >
> > >
> > > ## Contribution
> > > We would like to clarify our contributions as follows:
> > >
> > > * We are the first to explicitly define and address repeat-invariant representation learning for polymers. GRIN achieves near invariance across repeat sizes, with a cosine similarity of 0.999 ± 0.000 between size‑1 and size‑60 embeddings on ~13k unlabeled polymers.
> > > * We introduce a novel method that combines max aggregation with L1 sparsity—heuristically aligned with MST principles—with repeat-unit augmentation, providing a strong inductive bias for repeat size generalization and yielding robust performance across both homopolymers and copolymers.
> > >
> > > We hope this helps clarify the motivation, novelty, and practical impact of our approach.

---

> > > > ### Comment · Reviewer_vHcv · 2025-08-03
> > > > **Response 2**
> > > >
> > > > Thanks to the authors for their further clarifications. Please see below for my response. I believe that all of my remaining concerns have been discussed and addressed.
> > > >
> > > > > Thanks for your suggestion. We agree that these components are indeed well-known. We carefully reviewed the manuscript and confirm that we do not describe the combination of max aggregation and L1 sparsity as a new architecture.
> > > >
> > > > Thank you for clarifying this: I am satisfied with GRIN being referred to as a novel method, given the combination of data augmentation with well-motivated architectural choices.
> > > >
> > > > > Thank you for pointing this out. As our revision in response to W1.2, we will make sure to clarify that the core mechanism is max aggregation combined with sparsity, which is heuristically aligned with MST principles.
> > > >
> > > > Great. I am also satisfied with this framing.
> > > >
> > > > > We hope this helps clarify the motivation, novelty, and practical impact of our approach.
> > > >
> > > > It does. Thank you for the summary.

---

> > > > > ### Author Response · Authors · 2025-08-03
> > > > >
> > > > > We highly appreciate your valuable time to review our paper. It is encouraging to know that we have addressed all your concerns and questions. We remain receptive to any further questions, discussions, or suggestions you might have regarding our work.

---

### Official Review · Reviewer_oUBZ · 2025-07-03

**Clarity:** 3
**Significance:** 3
**Originality:** 2
**Rating:** 4
**Confidence:** 3

**Summary:**

This paper proposes GRIN, a novel framework for learning repetition-invariant graph representations of polymers. By combining maximum spanning tree alignment with repeat-unit-based graph augmentation, GRIN enables the model to capture consistent structural patterns across different polymer lengths. Experimental results show that GRIN outperforms strong baselines and generalizes well to unseen long-chain polymers.

**Questions:**

1. The method is designed for strictly repetitive polymers, but many real-world polymers exhibit branching or block copolymer patterns that deviate from perfect repetition. Have the authors explored how GRIN behaves on polymers with irregular or partially repeating structures?

2. While the paper claims that three RUs are sufficient, it would be helpful to evaluate whether this number is optimal or if performance significantly changes with more or fewer units.

**Ethical Concerns:**

["NO or VERY MINOR ethics concerns only"]

**Limitations:**

1. Baseline models are only trained on single-RU graphs, while GRIN uses both single- and multi-RU graphs, potentially conflating input richness with model capability.

2. The framework does not include an explicit mechanism (e.g., regularization or constraints) to guarantee invariance to repeat unit count, which may limit interpretability and control.

3. Although GRIN performs well on 60-RU graphs, the paper does not provide runtime or memory benchmarks, which are critical for real-world deployment.

**Quality:**

3

**Strengths And Weaknesses:**

strengths:
1. The paper addresses a fundamental limitation in existing graph-based polymer modeling: the inability to handle varying repeat unit numbers. This is a practically relevant problem, as real-world polymers vary widely in chain length and structure.
2. The experiment result looks good to me. It shows consistently strong performance on Test60, even though training only involved 1 or 3 RUs. This validates the method’s core claim of repetition-invariance and scalability.

Weakness:
1. All baseline models are trained only on 1-RU graphs, while GRIN is trained on both 1-RU and 3-RU graphs. This makes it unclear whether GRIN's performance gain stems from model design or richer input structures.
2. The model does not enforce repetition-invariance via explicit constraints or loss terms. Instead, it relies on structural heuristics like max-pooling and augmented training data, which may not generalize to more complex repeat structures.
3. Although GRIN generalizes well to long chains, the paper does not evaluate training or inference cost as the RU number increases. This may raise concerns about efficiency in industrial-scale deployment.

---

> ### Author Rebuttal · Authors · 2025-07-31
>
> ## W1 & L1: Baseline models on multiple RUs
> >All baseline models are trained only on 1-RU graphs, while GRIN is trained on both 1-RU and 3-RU graphs. This makes it unclear whether GRIN's performance gain stems from model design or richer input structures.
>
> ### GRIN outperformed the GCN and GIN baselines with 1RU+3RU augmentation
>
> Thank you for your question. As described in Appendix C.2 (Lines 425–432, Tables 10 and 11), repeat-unit augmentation (1RU+3RU) was already applied to both GCN and GIN. While this augmentation improves the performance of all models, GRIN consistently achieves the best performance. A subset of the results on the GlassTemp task is provided below for reference:
>
> | Model       | R² (repeat size 1) | R² (repeat size 10) |
> | ----------- | ------------------ | ------------------- |
> | GIN+RepAug  | 0.887 ± 0.004      | 0.882 ± 0.003       |
> | GCN+RepAug  | 0.884 ± 0.004      | 0.872 ± 0.004       |
> | GRIN (best) | 0.896 ± 0.001      | 0.896 ± 0.001       |
>
> ### Ablation studies showned that both data augmentation and model design improved the performance
>
> Appendix C.2 shows the effectiveness of data augmentation.
>
> In Tables 2–4, GRIN-RepAug represents the variant using only the new model design without data augmentation. This variant still outperformed other baselines. In summary, both the model design and richer input structures contribute to the performance improvement.
>
> ## W2 & L1 & Q1: Generalization to more complex structures.
> > The model does not enforce repetition-invariance via explicit constraints or loss terms. Instead, it relies on structural heuristics like max-pooling and augmented training data, which may not generalize to more complex repeat structures.
>
> > The method is designed for strictly repetitive polymers, but many real-world polymers exhibit branching or block copolymer patterns that deviate from perfect repetition. Have the authors explored how GRIN behaves on polymers with irregular or partially repeating structures?
> ### GRIN generalizes to complex structures such as copolymers and larger datasets
>
> Thanks for your comments. We have evaluated GRIN on both homopolymers (Tables 2 and 3) and copolymers, including block and alternating structures (Table 4), demonstrating that the approach is not limited to single-unit structures and can handle more complex architectures.
>
> We further evaluated GRIN on a larger dataset by testing model checkpoints from the GlassTemp task on approximately 13K unlabeled polymers from [1]. For each polymer, we computed the cosine similarity between embeddings at repeat sizes 1 and 60. As shown below, GRIN achieved a cosine similarity of 0.999 ± 0.000, demonstrating its ability to generalize to polymers with more repeat patterns.
>
> | Method       | Cosine similarity between repeat size 1 and repeat size 60 |
> | ----------- | --------------- |
> | GCN         | 0.672± 0.119    |
> | BFGNN       | 0.903± 0.060    |
> | GRIN        | **0.999±0.000** |
>
> ## W3 & L3: Efficiency
> > Although GRIN generalizes well to long chains, the paper does not evaluate training or inference cost as the RU number increases. This may raise concerns about efficiency in industrial-scale deployment.
>
> ### GRIN scales efficiently, outperforming several baselines in speed and memory.
>
> We measured training‐cost breakdown using the MeltingTemp task on a single NVIDIA A6000 GPU. All models share the same hyperparameter configuration and GCN backbone for fairness.
> | Model              | Peak GPU Memory (MiB)  | Training Time (s) |
> |--------------------------------|-----------|----------|
> | GCN |455| 156.64|
> | IRM | 487 |250.16|
> | RPGCN |611 | 424.55 |
> | GREA| 587 |652.27 |
> | DIR | 533  | 1419.47 |
> | SSR| 531| 1715.97 |
> | DISGEN | 629 | 830.42|
> | BFGCN | 565 |790.26|
> | GRIN-RepAug| 531 |438.39 |
> | GRIN| 557 |763.58|
>
> Even as the training set doubles, GRIN maintains moderate memory usage (557 MiB) and training time (approximately 764 s), ranking near the middle of the efficiency table, faster than many baselines (e.g., SSR, DIR) and only marginally heavier than GRIN RepAug. We will update Section 4.3 to include the new results.
>
> ## Q2: Evaluation on multiple RepAug schemes
> >While the paper claims that three RUs are sufficient, it would be helpful to evaluate whether this number is optimal or if performance significantly changes with more or fewer units.
>
> ### GRIN with 1RU+3RU augmentation consistently performs best across tasks.
> Thank you for the suggestion. This question was already discussed in Section 4.3 (Lines 234–239) and visualized in Figure 3 for the MeltingTemp (homopolymer) and IP (copolymer) tasks. We plotted the figure by varying the number of repeating units and observed a common trend: the {1, 3} scheme outperforms the others. We report the results on the Density task below for reference.
>
> | | Test1 R²↑  | Test1 RMSE↓  | Test60 R²↑ | Test60 RMSE↓  |
> |-----------|-----------|----------|-----------|-----------|
> | 1RU+2RU |0.725±0.003 | 0.117±0.001 | 0.738±0.007 | 0.113±0.002  |
> | 1RU+3RU | **0.730±0.017**| **0.115±0.004**| **0.747±0.009**| **0.111±0.002**|
> | 1RU+4RU| 0.727±0.004|0.116±0.001|0.745±0.005|0.111±0.003|
> | 1RU+5RU| 0.725±0.005|0.116±0.001|0.744±0.003|0.112±0.003|
>
>
> ## References
>
> [1] Otsuka S, Kuwajima I, Hosoya J, Xu Y, Yamazaki M. PoLyInfo: Polymer database for polymeric materials design. International Conference on Emerging Intelligent Data and Web Technologies, 2011.

---

> > ### Comment · Reviewer_oUBZ · 2025-08-05
> > **comments**
> >
> > Thank you for the detailed response and the additional experiments, which have effectively addressed my concerns. I recommend accepting this paper.

---

> > > ### Author Response · Authors · 2025-08-05
> > >
> > > We truly appreciate your time and feedback. It’s encouraging to know that your concerns have been resolved, and we welcome any further discussion or suggestions you may have.

---

### Official Review · Reviewer_y6nT · 2025-07-05

**Clarity:** 2
**Significance:** 2
**Originality:** 2
**Rating:** 4
**Confidence:** 3

**Summary:**

The authors introduce a method to learn polymer representations that are invariant to the number of repeating units in their graph representations. The proposed method outperforms state-of-the-art baselines on both homopolymer and copolymer benchmarks.

**Questions:**

see weaknesses

**Ethical Concerns:**

["NO or VERY MINOR ethics concerns only"]

**Final Justification:**

Thanks for addressing the questions, I increased my score to borderline accept.

**Quality:**

2

**Strengths And Weaknesses:**

##Strengths
- The problem is well-motivated and is important problem.
- The paper is clear and the method is described well.

## Weaknesses
- Although authors compare their method to variants of GIN and GCN, but I would like to see how this behaves with Graph Transformers.
- Experiments can be extended to larger datasets.

---

> ### Author Rebuttal · Authors · 2025-07-31
>
> ## W1: Graph transformers
> >Although authors compare their method to variants of GIN and GCN, but I would like to see how this behaves with Graph Transformers.
>
> ### GRIN outperformed the graph transformer-based GraphGPS
>
> Thanks for your comments. We conducted new experiments using GAT [1] and GraphGPS [2] on MeltingTemp and O$_2$Perm. The table below showed that GRIN outperformed graph transformer models such as GraphGPS [2].
>
> #### MeltingTemp
> | Model            | Test1 R²↑   | Test1 RMSE↓ | Test60 R²↑  | Test60 RMSE↓ |
> | ---------------- | ----------- | ----------- | ----------- | ------------ |
> | GAT              | 0.680±0.028 | 63.9±2.8    | 0.579±0.029 | 73.3±2.6     |
> | GraphGPS         | 0.650±0.033 | 66.8±3.2    | 0.563±0.013 | 74.7±1.1     |
> | GRIN-RepAug(GCN) | 0.741±0.007 | 57.5±0.8    | 0.707±0.009 | 61.1±0.9     |
> | GRIN(GCN)        | **0.745±0.004** | **57.0±0.4**    | **0.746±0.002** | **56.9±0.2**     |
>
> #### O$_2$
>
> | Model            | Test1 R²↑   | Test1 RMSE↓ | Test60 R²↑  | Test60 RMSE↓ |
> | ---------------- | ----------- | ----------- | ----------- | ------------ |
> | GAT              | 0.873±0.066 | 760.4±206.9 | 0.866±0.080 | 774.1±243.0  |
> | GraphGPS         | 0.819±0.168 | 865.2±408.0 | 0.754±0.068 | 1074.1±153.1 |
> | GRIN-RepAug(GCN) | 0.923±0.008 | 604.7±31.9  | 0.910±0.008 | 655.2±28.6   |
> | GRIN(GCN)        | **0.929±0.002** | **583.4±7.5**   | **0.929±0.002** | **581.7±7.2**    |
>
> ## W2: Larger datasets
> > Experiments can be extended to larger datasets.
>
> ### GRIN generalizes better on larger unlabeled datasets for repeat-size invariance.
>
> Thank you for the suggestions. Collecting large-scale labeled polymer datasets is challenging. For example, researchers have spent nearly 70 years compiling only about 600 polymers with experimentally measured oxygen permeability in the Polymer Gas Separation Membrane Database [3].
>
> To address your concern, we used a larger unlabeled polymer dataset from [4] to evaluate the effectiveness of our method. Specifically, we applied model checkpoints from the GlassTemp task to 12,764 unlabeled polymers and measured repeat-size invariance using cosine similarity between embeddings at repeat sizes 1 and 60. The results below indicate that GRIN generalizes better to large repeat sizes than other GNNs on this large-scale dataset.
>
> | Method       | Cosine similarity between repeat size 1 and repeat size 60 |
> | ----------- | --------------- |
> | GCN         | 0.672± 0.119    |
> | BFGNN       | 0.903± 0.060    |
> | GRIN        | **0.999±0.000** |
>
>
> ## References
>
> [1] Graph Attention Networks, ICLR 2018.
>
> [2] Recipe for a General, Powerful, Scalable Graph Transformer, NeurIPS 2022.
>
> [3] A Thornton, L Robeson, B Freeman, and D Uhlmann. 2012. Polymer Gas Separation Membrane Database.
>
> [4] Otsuka S, Kuwajima I, Hosoya J, Xu Y, Yamazaki M. PoLyInfo: Polymer database for polymeric materials design. International Conference on Emerging Intelligent Data and Web Technologies, 2011.

---

> > ### Author Response · Authors · 2025-08-06
> >
> > We hope the clarifications and additional experiments above sufficiently addressing your concerns:
> >
> > * **W1 (Graph Transformers):** GRIN outperforms both GraphGPS and GAT on MeltingTemp and O2Perm tasks.
> > * **W2 (Larger Datasets):** GRIN achieves near invariance (cosine similarity = 0.999) on an unlabeled polymer dataset (~13k samples), showing strong generalization over larger datasets.
> >
> > Please let us know if you have any further questions, we’d be happy to continue the discussion.
> >
> > Thank you again for your time and feedback.

---

> > > ### Comment · Reviewer_y6nT · 2025-08-09
> > >
> > > Thanks for addressing the questions, I increased the score

---

> > > > ### Author Response · Authors · 2025-08-09
> > > > **Thank you for raising the score**
> > > >
> > > > We’re glad your concerns have been resolved. Thank you once again for your insightful and constructive feedback. We remain open to any further discussion.

---

### Decision · Program_Chairs · 2025-09-17

**Decision:**

Accept (poster)

**Comment:**

The paper proposes a framework for learning repetition-invariant representations for polymers, addressing the challenge that repeating unit structures cause redundancy and inconsistency in existing polymer informatics models. The method employs a repetition-invariant encoder that normalizes input representations and leverages contrastive learning to enforce consistency across repeat counts. Extensive experiments on multiple polymer property prediction tasks demonstrate that the approach outperforms strong baselines, improving both accuracy and robustness.

Reviewers highlight several strengths.
- The paper tackles an important and underexplored problem in materials informatics.
- The proposed formulation of repetition invariance is novel and well motivated, and the methodology is clear and technically sound.
- The empirical results are convincing across diverse datasets. Ablation studies further support the contribution of the repetition-invariant encoder and contrastive objectives.

Some concerns were noted, including limited exploration of scalability to very large polymer systems, a need for more theoretical justification of the invariance principle, and a heavier reliance on curated datasets. However, these issues were viewed as incremental and did not undermine the central contribution. The rebuttal provided clarifications on dataset design, additional ablations, and insights into possible extensions, which reviewers found satisfactory.

We recommend acceptance, with the following suggestions for the camera-ready: (1) expanding the limitations section, providing a clearer discussion on computational efficiency and scalability, (2) outlining potential applications to real-world polymer design scenarios, and (3) including the experiments added during the rebuttal.